# From Bayesian Sparsity to Gated Recurrent Nets

**Hao He**
Massachusetts Institute of Technology
haohe@mit.edu

**Bo Xin**
Microsoft Research, Beijing, China
jimxinbo@gmail.com

**Satoshi Ikehata**
National Institute of Informatics
satoshi.ikehata@gmail.com

**David Wipf**
Microsoft Research, Beijing, China
davidwipf@gmail.com

## Abstract

The iterations of many first-order algorithms, when applied to minimizing common regularized regression functions, often resemble neural network layers with pre-specified weights. This observation has prompted the development of learning-based approaches that purport to replace these iterations with enhanced surrogates forged as DNN models from available training data. For example, important NP-hard sparse estimation problems have recently benefitted from this genre of upgrade, with simple feedforward or recurrent networks ousting proximal gradient-based iterations. Analogously, this paper demonstrates that more powerful Bayesian algorithms for promoting sparsity, which rely on complex multi-loop majorization-minimization techniques, mirror the structure of more sophisticated long short-term memory (LSTM) networks, or alternative gated feedback networks previously designed for sequence prediction. As part of this development, we examine the parallels between latent variable trajectories operating across multiple time-scales during optimization, and the activations within deep network structures designed to adaptively model such characteristic sequences. The resulting insights lead to a novel sparse estimation system that, when granted training data, can estimate optimal solutions efficiently in regimes where other algorithms fail, including practical direction-of-arrival (DOA) and 3D geometry recovery problems. The underlying principles we expose are also suggestive of a learning process for a richer class of multi-loop algorithms in other domains.

## 1 Introduction

Many practical iterative algorithms for minimizing an energy function $\mathcal{L}_y(\boldsymbol{x})$, parameterized by some vector $\boldsymbol{y}$, adopt the updating prescription

$$\boldsymbol{x}^{(t+1)} = f(\boldsymbol{A}\boldsymbol{x}^{(t)} + \boldsymbol{B}\boldsymbol{y}), \tag{1}$$

where $t$ is the iteration count, $\boldsymbol{A}$ and $\boldsymbol{B}$ are fixed matrices/filters, and $f$ is a point-wise nonlinear operator. When we treat $\boldsymbol{B}\boldsymbol{y}$ as a bias or exogenous input, then the progression of these iterations through time resembles activations passing through the layers (indexed by $t$) of a deep neural network (DNN) [20, 30, 34, 38]. It then naturally begs the question: If we have access to an ensemble of pairs $\{\boldsymbol{y}, \boldsymbol{x}^*\}$, where $\boldsymbol{x}^* = \arg\min_{\boldsymbol{x}} \mathcal{L}_y(\boldsymbol{x})$, can we train an appropriately structured DNN to produce a minimum of $\mathcal{L}_y(\boldsymbol{x})$ when presented with an arbitrary new $\boldsymbol{y}$ as input? If $\boldsymbol{A}$ and $\boldsymbol{B}$ are fixed for all $t$, this process can be interpreted as training a recurrent neural network (RNN), while if they vary, a deep feedforward network with independent weights on each layer is a more apt description.

Although many of our conclusions may ultimately have broader implications, in this work we focus on minimizing the ubiquitous sparse estimation problem

$$\mathcal{L}_y(\boldsymbol{x}) = \|\boldsymbol{y} - \boldsymbol{\Phi}\boldsymbol{x}\|_2^2 + \lambda\|\boldsymbol{x}\|_0, \qquad (2)$$

where $\boldsymbol{\Phi} \in \mathbb{R}^{n \times m}$ is an overcomplete matrix of feature vectors, $\|\cdot\|_0$ is the $\ell_0$ norm equal to a count of the nonzero elements in a vector, and $\lambda > 0$ is a trade-off parameter. Although crucial to many applications [2, 9, 13, 17, 23, 27], solving (2) is NP-hard, and therefore efficient approximations are sought. Popular examples with varying degrees of computational overhead include convex relaxations such as $\ell_1$-norm regularization [4, 8, 32] and many flavors of iterative hard-thresholding (IHT) [5, 6].

In most cases, these approximate algorithms can be implemented via (1), where $\boldsymbol{A}$ and $\boldsymbol{B}$ are functions of $\boldsymbol{\Phi}$, and the nonlinearity $f$ is, for example, a hard-thresholding operator for IHT or soft-thresholding for convex relaxations. However, the Achilles' heel of all these approaches is that they will generally not converge to good approximate minimizers of (2) if $\boldsymbol{\Phi}$ has columns with a high degree of correlation [5, 8], which is unfortunately often the case in practice [35].

To mitigate the effects of such correlations, we could leverage the aforementioned correspondence with common DNN structures to learn something like a correlation-invariant algorithm or update rules [38], although in this scenario our starting point would be an algorithmic format with known deficiencies. But if our ultimate goal is to learn a new sparse estimation algorithm that efficiently compensates for structure in $\boldsymbol{\Phi}$, then it seems reasonable to invoke iterative algorithms known *a priori* to handle such correlations directly as our template for learned network layers. One important example is sparse Bayesian learning (SBL) [33], which has been shown to solve (2) using a principled, multi-loop majorization-minimization approach [22] even in cases where $\boldsymbol{\Phi}$ displays strong correlations [35]. Herein we demonstrate that, when judiciously unfolded, SBL iterations can be formed into variants of long short-term memory (LSTM) cells, one of the more popular recurrent deep neural network architectures [21], or gated extensions thereof [12]. The resulting network dramatically outperforms existing methods in solving (2) with a minimal computational budget. Our high-level contributions can be summarized as follows:

- Quite surprisingly, we demonstrate that the SBL objective, which explicitly compensates for correlated dictionaries, can be optimized using iteration structures that map directly to popular LSTM cells despite its radically different origin. This association significantly broadens recent work connecting elementary, one-step iterative sparsity algorithms like (1) with simple recurrent or feedforward deep network architectures [20, 30, 34, 38].

- At its core, any SBL algorithm requires coordinating inner- and outer-loop computations that produce expensive latent posterior variances (or related, derived quantities) and optimized coefficient estimates respectively. Although this process can in principle be accommodated via canonical LSTM cells, such an implementation will enforce that computation of latent variables rigidly map to predefined subnetworks corresponding with various gating structures, ultimately administering a fixed schedule of switching between loops. To provide greater flexibility in coordinating inner- and outer-loops, we propose a richer gated-feedback LSTM structure for sparse estimation.

- We achieve state-of-the-art performance on several empirical tasks, including direction-of-arrival (DOA) estimation [28] and 3D geometry recovery via photometric stereo [37]. In these and other cases, our approach produces higher accuracy estimates at a fraction of the computational budget. These results are facilitated by a novel online data generation process.

- Although learning-to-learn style approaches [1, 20, 30, 34] have been commonly applied to relatively simple gradient descent optimization templates, this is the first successful attempt we are aware of to learn a complex, multi-loop, majorization-minimization algorithm [22]. We envision that such a strategy can have wide-ranging implications beyond the sparse estimation problems explored herein given that it is often not obvious how to optimally tune loop execution to balance both complexity and estimation accuracy in practice.

## 2 Connecting SBL and LSTM Networks

This section first reviews the basic SBL model, followed an algorithmic characterization of how correlation structure can be handled during sparse estimation. Later we derive specialized SBL update rules that reveal a close association with LSTM cells.

## 2.1 Original SBL Model

Given an observed vector $\boldsymbol{y} \in \mathbb{R}^n$ and feature dictionary $\boldsymbol{\Phi} \in \mathbb{R}^{n \times m}$, SBL assumes the Gaussian likelihood model and a parameterized zero-mean Gaussian prior for the unknown coefficients $\boldsymbol{x} \in \mathbb{R}^m$ given by

$$p(\boldsymbol{y}|\boldsymbol{x}) \propto \exp\left[-\tfrac{1}{2\lambda}\|\boldsymbol{y} - \boldsymbol{\Phi x}\|_2^2\right], \quad \text{and} \quad p(\boldsymbol{x};\boldsymbol{\gamma}) \propto \exp\left[-\tfrac{1}{2}\boldsymbol{x}^\top \boldsymbol{\Gamma}^{-1} \boldsymbol{x}\right], \quad \boldsymbol{\Gamma} \triangleq \text{diag}[\boldsymbol{\gamma}] \quad (3)$$

where $\lambda > 0$ is a fixed variance factor and $\boldsymbol{\gamma}$ denotes a vector of unknown hyperparamters [33]. Because both likelihood and prior are Gaussian, the posterior $p(\boldsymbol{x}|\boldsymbol{y};\boldsymbol{\gamma})$ is also Gaussian, with mean $\hat{\boldsymbol{x}}$ satisfying

$$\hat{\boldsymbol{x}} = \boldsymbol{\Gamma \Phi}^\top \boldsymbol{\Sigma}_y^{-1} \boldsymbol{y}, \quad \text{with } \boldsymbol{\Sigma}_y \triangleq \boldsymbol{\Phi \Gamma \Phi}^\top + \lambda \boldsymbol{I}. \quad (4)$$

Given the lefthand-side multiplication by $\boldsymbol{\Gamma}$ in (4), $\hat{\boldsymbol{x}}$ will have a matching sparsity profile or support pattern as $\boldsymbol{\gamma}$, meaning that the locations of zero-valued elements will align or $\text{supp}[\hat{\boldsymbol{x}}] = \text{supp}[\boldsymbol{\gamma}]$. Ultimately then, the SBL strategy shifts from directly searching for some optimally sparse $\hat{\boldsymbol{x}}$, to an optimally sparse $\boldsymbol{\gamma}$. For this purpose we marginalize over $\boldsymbol{x}$ (treating it initially as hidden or nuisance data) and then maximize the resulting type-II likelihood function with respect to $\boldsymbol{\gamma}$ [26]. Conveniently, the resulting convolution-of-Gaussians integral is available in closed-form [33] such that we can equivalently minimize the negative log-likelihood

$$\mathcal{L}(\boldsymbol{\gamma}) \;=\; -\log \int p(\boldsymbol{y}|\boldsymbol{x})p(\boldsymbol{x};\boldsymbol{\gamma})d\boldsymbol{x} \;\equiv\; \boldsymbol{y}^\top \boldsymbol{\Sigma}_y^{-1} \boldsymbol{y} + \log|\boldsymbol{\Sigma}_y|. \quad (5)$$

Given an optimal $\boldsymbol{\gamma}$ so obtained, we can compute the posterior mean estimator $\hat{\boldsymbol{x}}$ via (4). Equivalently, this same posterior mean estimator can be obtained by an iterative reweighted $\ell_1$ process described next that exposes subtle yet potent sparsity-promotion mechanisms.

## 2.2 Iterative Reweighted $\ell_1$ Implementation

Although not originally derived this way, SBL can be implemented using a modified form of iterative reweighted $\ell_1$-norm optimization that exposes its agency for producing sparse estimates. In general, if we replace the $\ell_0$ norm from (2) with any smooth approximation $g(|\boldsymbol{x}|)$, where $g$ is a concave, non-decreasing function and $|\cdot|$ applies elementwise, then cost function descent[1] can be guaranteed using iterations of the form [36]

$$\boldsymbol{x}^{(t+1)} \leftarrow \arg\min_{\boldsymbol{x}} \tfrac{1}{2}\|\boldsymbol{y} - \boldsymbol{\Phi x}\|_2^2 + \lambda \sum_i w_i^{(t)} |x_i|, \qquad w_i^{(t+1)} \leftarrow \partial g(\boldsymbol{u})/\partial u_i \big|_{u_i = \left|x_i^{(t+1)}\right|}, \quad \forall i. \quad (6)$$

This process can be viewed as a multi-loop, majorization-minimization algorithm [22] (a generalization of the EM algorithm [15]), whereby the inner-loop involves computing $\boldsymbol{x}^{(t+1)}$ by minimizing a first-order, upper-bounding approximation $\|\boldsymbol{y} - \boldsymbol{\Phi x}\|_2^2 + \lambda \sum_i w_i^{(t)} |x_i|$, while the outer-loop updates the bound/majorizer itself as parameterized by the weights $\boldsymbol{w}^{(t+1)}$. Obviously, if $g(\boldsymbol{u}) = \boldsymbol{u}$, then $\boldsymbol{w}^{(t)} = \boldsymbol{1}$ for all $t$, and (6) reduces to the Lasso objective for $\ell_1$ norm regularized sparse regression [32], and only a single iteration is required. However, one popular non-trivial instantiation of this approach assumes $g(\boldsymbol{u}) = \sum_i \log(u_i + \epsilon)$ with $\epsilon > 0$ a user-defined parameter [10]. The corresponding weights then become $w_i^{(t+1)} = \left(\left|x_i^{(t+1)}\right| + \epsilon\right)^{-1}$, and we observe that once any particular $x_i^{(t+1)}$ becomes large, the corresponding weight becomes small and at the next iteration a weaker penalty will be applied. This prevents the overshrinkage of large coefficients, a well-known criticism of $\ell_1$ norm penalties [16].

In the context of SBL, there is no closed-form $w_i^{(t+1)}$ update except in special cases. However, if we allow for additional latent structure, which we later show is akin to the memory unit of LSTM cells, a viable recurrency emerges for computing these weights and elucidating their effectiveness in dealing with correlated dictionaries. In particular we have:

**Proposition 1.** *If weights $\boldsymbol{w}^{(t+1)}$ satisfy*

$$\left(w_i^{(t+1)}\right)^2 = \min_{\boldsymbol{z}:supp[\boldsymbol{z}] \subseteq supp[\boldsymbol{\gamma}^{(t)}]} \frac{1}{\lambda}\|\boldsymbol{\phi}_i - \boldsymbol{\Phi z}\|_2^2 + \sum_{j \in supp[\boldsymbol{\gamma}^{(t)}]} \frac{z_j^2}{\gamma_j^{(t+1)}} \quad (7)$$

*for all $i$, then the iterations (6), with $\gamma_j^{(t+1)} = \left[ w_j^{(t)} \right]^{-1} \left| x_j^{(t+1)} \right|$, are guaranteed to reduce or leave unchanged the SBL objective (5). Also, at each iteration, $\boldsymbol{\gamma}^{(t+1)}$ and $\boldsymbol{x}^{(t+1)}$ will satisfy (4).*

Unlike the traditional sparsity penalty mentioned above, with SBL we see that the $i$-th weight $w_i^{(t+1)}$ is not dependent solely on the value of the $i$-th coefficient $x_i^{(t+1)}$, but rather on *all* the latent hyperparameters $\boldsymbol{\gamma}^{(t+1)}$ and therefore ultimately prior-iteration weights $\boldsymbol{w}^{(t)}$ as well. Moreover, because the fate of each sparse coefficient is linked together, correlation structure can be properly accounted for in a progressive fashion.

More concretely, from (7) it is immediately apparent that if $\boldsymbol{\phi}_i \approx \boldsymbol{\phi}_{i'}$ for some indeces $i$ and $i'$ (meaning a large degree of correlation), then it is highly likely that $w_i^{(t+1)} \approx w_{i'}^{(t+1)}$. This is simply because the regularized residual error that emerges from solving (7) will tend to be quite similar when $\boldsymbol{\phi}_i \approx \boldsymbol{\phi}_{i'}$. In this situation, a suboptimal solution will not be prematurely enforced by weights with large, spurious variance across a correlated group of basis vectors. Instead, weights will differ substantially only when the corresponding columns have meaningful differences relative to the dictionary as a whole, in which case such differences can help to avoid overshrinkage as before.

A crucial exception to this perspective occurs when $\boldsymbol{\gamma}^{(t+1)}$ is highly sparse, or nearly so, in which case there are limited degrees of freedom with which to model even small differences between some $\boldsymbol{\phi}_i$ and $\boldsymbol{\phi}_{i'}$. However, such cases can generally only occur when we are in the neighborhood of ideal, maximally sparse solutions by definition [35], when different weights are actually desirable even among correlated columns for resolving the final sparse estimates.

## 2.3 Revised SBL Iterations

Although presumably there are multiple ways such an architecture could be developed, in this section we derive specialized SBL iterations that will directly map to one of the most common RNN structures, namely LSTM networks. With this in mind, the notation we adopt has been intentionally chosen to facilitate later association with LSTM cells. We first define

$$\boldsymbol{w}^{(t)} \triangleq \mathrm{diag} \left[ \boldsymbol{\Phi}^\top \left( \lambda \boldsymbol{I} + \boldsymbol{\Phi} \boldsymbol{\Gamma}^{(t)} \boldsymbol{\Phi}^\top \right)^{-1} \boldsymbol{\Phi} \right]^{\frac{1}{2}} \quad \text{and} \quad \boldsymbol{\nu}^{(t)} \triangleq \boldsymbol{u}^{(t)} + \mu \boldsymbol{\Phi}^\top \left( \boldsymbol{y} - \boldsymbol{\Phi} \boldsymbol{u}^{(t)} \right), \quad (8)$$

where $\boldsymbol{\Gamma}^{(t)} \triangleq \mathrm{diag} \left[ \boldsymbol{\gamma}^{(t)} \right]$, $\boldsymbol{u}^{(t)} \triangleq \boldsymbol{\Gamma}^{(t)} \boldsymbol{\Phi}^\top \left( \lambda \boldsymbol{I} + \boldsymbol{\Phi} \boldsymbol{\Gamma}^{(t)} \boldsymbol{\Phi}^\top \right)^{-1} \boldsymbol{y}$, and $\mu > 0$ is a constant. As will be discussed further below, $\boldsymbol{w}^{(t)}$ serves the exact same role as the weights from (7), hence the identical notation. We then partition our revised SBL iterations as so-called *gate* updates

$$\boldsymbol{\sigma}_{in}^{(t)} \leftarrow \left[ \boldsymbol{\alpha} \left( \boldsymbol{\gamma}^{(t)} \right) \odot \left( \left| \boldsymbol{\nu}^{(t)} \right| - 2\lambda \boldsymbol{w}^{(t)} \right) \right]_+, \qquad \boldsymbol{\sigma}_f^{(t)} \leftarrow \boldsymbol{\beta} \left( \boldsymbol{\gamma}^{(t)} \right), \qquad \boldsymbol{\sigma}_{out}^{(t)} \leftarrow \left( \boldsymbol{w}^{(t)} \right)^{-1}, \quad (9)$$

*cell* updates

$$\bar{\boldsymbol{x}}^{(t+1)} \leftarrow \mathrm{sign} \left[ \boldsymbol{\nu}^{(t)} \right], \qquad \boldsymbol{x}^{(t+1)} \leftarrow \boldsymbol{\sigma}_f^{(t)} \odot \boldsymbol{x}^{(t)} + \boldsymbol{\sigma}_{in}^{(t)} \odot \bar{\boldsymbol{x}}^{(t+1)}, \quad (10)$$

and *output* updates

$$\boldsymbol{\gamma}^{(t+1)} \quad \leftarrow \quad \boldsymbol{\sigma}_{out}^{(t)} \odot \left| \boldsymbol{x}^{(t+1)} \right|, \quad (11)$$

where the inverse and absolute-value operators are applied element-wise when a vector is the argument, and at least for now, $\boldsymbol{\alpha}$ and $\boldsymbol{\beta}$ define arbitrary functions. Moreover, $\odot$ denotes the Hadamard product and $[\cdot]_+$ sets negative values to zero and leaves positive quantities unchanged, also in an element-wise fashion, i.e., it acts just like a rectilinear (ReLU) unit [29]. Note also that the gate and cell updates in isolation can be viewed as computing a first-order, partial solution to the inner-loop weighted $\ell_1$ optimization problem from (6).

Starting from some initial $\boldsymbol{\gamma}^{(0)}$ and $\boldsymbol{x}^{(0)}$, we will demonstrate in the next section that these computations closely mirror a canonical LSTM network unfolded in time with $\boldsymbol{y}$ acting as a constant input applied at each step. Before doing so however, we must first demonstrate that (8)−(11) indeed serve to reduce the SBL objective. For this purpose we require the following definition:

**Definition 2.** *We say that the iterations (8)−(11) satisfy the monotone cell update property if*

$$\|\boldsymbol{y} - \boldsymbol{\Phi}\boldsymbol{u}^{(t)}\|_2^2 + 2\lambda \sum_i w_i^{(t)}|u_i^{(t)}| \;\geq\; \|\boldsymbol{y} - \boldsymbol{\Phi}\boldsymbol{x}^{(t+1)}\|_2^2 + 2\lambda \sum_i w_i^{(t)}|x_i^{(t+1)}|, \;\; \forall t. \qquad (12)$$

Note that for rather inconsequential technical reasons this definition involves $\boldsymbol{u}^{(t)}$, which can be viewed as a proxy for $\boldsymbol{x}^{(t)}$. We then have the following:

**Proposition 3.** *The iterations (8)−(11) will reduce or leave unchanged (5) for all $t$ provided that $\mu \in \left(0, \lambda/\left\|\boldsymbol{\Phi}^\top\boldsymbol{\Phi}\right\|\right)$ and $\boldsymbol{\alpha}$ and $\boldsymbol{\beta}$ are chosen such that the monotone cell update property holds.*

In practical terms, the simple selections $\boldsymbol{\alpha}(\boldsymbol{\gamma}) = \mathbf{1}$ and $\boldsymbol{\beta}(\boldsymbol{\gamma}) = \mathbf{0}$ will provably satisfy the monotone cell update property (see proof details in the supplementary). However, for additional flexibility, $\boldsymbol{\alpha}$ and $\boldsymbol{\beta}$ could be selected to implement various forms of momentum, ultimately leading to cell updates akin to the popular FISTA [4] or monotonic FISTA [3] algorithms. In both cases, old values $\boldsymbol{x}^{(t)}$ are precisely mixed with new factors $\bar{\boldsymbol{x}}^{(t+1)}$ to speed convergence (in the present circumstances, $\boldsymbol{\sigma}_f^{(t)}$ and $\boldsymbol{\sigma}_{in}^{(t)}$ respectively modulate this mixing process via (10)). Of course the whole point of casting the SBL iterations as an RNN structure to begin with is so that we may ultimately *learn* these types of functions, without the need for hand-crafting suboptimal iterations up front.

### 2.4   Correspondences with LSTM Components

We will now flesh out how the SBL iterations presented in Section 2.3 display the same structure as a canonical LSTM cell, the only differences being the shape of the nonlinearities, and the exact details of the gate subnetworks. To facilitate this objective, Figure 1 contains a canonical LSTM network structure annotated with SBL-derived quantities. We now walk through these correspondences.

First, the exogenous input to the network is the observation vector $\boldsymbol{y}$, which does not change from time-step to time-step. This is much like the strategy used by feedback networks for obtaining incrementally refined representations [40]. The output at time-step $t$ is $\boldsymbol{\gamma}^{(t)}$, which serves as the current estimate of the SBL hyperparameters. In contrast, we treat $\boldsymbol{x}^{(t)}$ as the internal LSTM memory cell, or the latent cell state.[2] This deference to $\boldsymbol{\gamma}^{(t)}$ directly mirrors the emphasis SBL places on learning variances per the marginalized cost from (5) while treating $\boldsymbol{x}^{(t)}$ as hidden data, and in some sense flips the coefficient-centric script used in producing (6).[3]

Proceeding further, $\boldsymbol{\gamma}^{(t)}$ is fed to four separate layers/subnetworks (represented by yellow boxes in Figure 1): (i) the *forget* gate $\boldsymbol{\sigma}_f^{(t)}$, (ii) the *input* gate $\boldsymbol{\sigma}_{in}^{(t)}$, (iii) the *output* gate $\boldsymbol{\sigma}_{out}^{(t)}$, and (iv) the candidate input update $\bar{\boldsymbol{x}}^{(t)}$. The forget gate computes scaling factors for each element of $\boldsymbol{x}^{(t)}$, with small values of the gate output suggesting that we 'forget' the corresponding old cell state elements. Similarly the input gate determines how large we rescale signals from the candidate input update $\bar{\boldsymbol{x}}^{(t)}$. These two re-weighted quantities are then mixed together to form the new cell state $\boldsymbol{x}^{(t+1)}$. Finally, the output gate modulates how new $\boldsymbol{\gamma}^{(t+1)}$ are created as scaled versions of the updated cell state.

Regarding details of these four subnetworks, based on the update templates from (9) and (10), we immediately observe that the required quantities depend directly on (8). Fortunately, both $\boldsymbol{\nu}^{(t)}$ and $\boldsymbol{w}^{(t)}$ can be naturally computed using simple feedforward subnetwork structures.[4] These values can either be computed in full (ideal case), or partially to reduce the computational burden. In any event, once obtained, the respective gates and candidate cell input updates can be computed by applying final non-linearities. Note that $\boldsymbol{\alpha}$ and $\boldsymbol{\beta}$ are treated as arbitrary subnetwork structures at this point that can be learned.

A few cosmetic differences remain between this SBL implementation and a canonical LSTM network. First, the final non-linearity for LSTM gating subnetworks is often a sigmoidal activation, whereas SBL is flexible with the forget gate (via $\boldsymbol{\beta}$), while effectively using a ReLU unit for the input gate and an inverse function for the output gate. Moreover, for the candidate cell update subnetwork, SBL replaces the typical tanh nonlinearity with a quantized version, the sign function, and likewise, for the output nonlinearity an absolute value operator (abs) is used. Finally, in terms of internal subnetwork structure, there is some parameter sharing since $\boldsymbol{\sigma}_{in}^{(t)}$, $\boldsymbol{\sigma}_{out}^{(t)}$, and $\bar{\boldsymbol{x}}^{(t)}$ are connected via $\boldsymbol{\nu}^{(t)}$ and $\boldsymbol{w}^{(t)}$.

Of course in all cases we need not necessarily share parameters nor abide by these exact structures. In fact there is nothing inherently optimal about the particular choices used by SBL; rather it is merely that these structures happen to reproduce the successful, yet hand-crafted SBL iterations. But certainly there is potential in replacing such iterations with learned LSTM-like surrogates, at least when provided with access to sufficient training data as in prior attempts to learn sparse estimation algorithms [20, 34, 38].

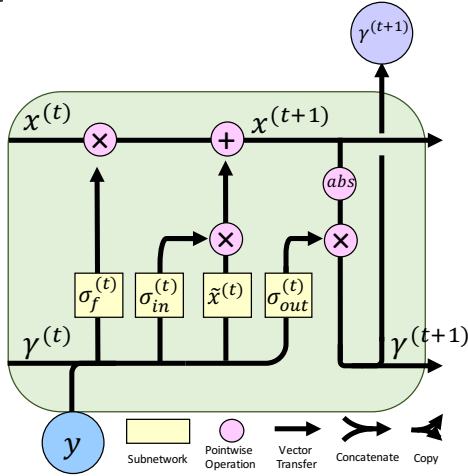

Figure 1: LSTM/SBL Network

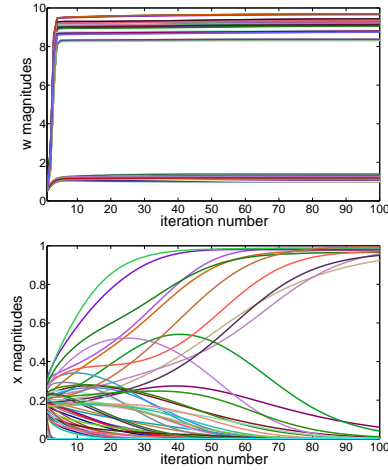

Figure 2: SBL Dynamics

## 3 Extension to Gated Feedback Networks

Although SBL iterations can be molded into an LSTM structure as we have shown, there remain hints that the full potential of this association may be presently undercooked. Here we first empirically examine the trajectories of SBL iterations produced via the LSTM-like rules derived in Section 2.3. This process will later serve to unmask certain characteristic dynamics operating across different time scales that are suggestive of a richer class of gated recurrent network structures inspired by sequence prediction tasks [12].

### 3.1 Trajectory Analysis of SBL Iterations

To begin, Figure 2 displays sample trajectories of $\boldsymbol{w}^{(t)} \in \mathbb{R}^{100}$ (top) and $\boldsymbol{x}^{(t)} \in \mathbb{R}^{100}$ (bottom) during execution of (8)−(11) on a simple representative problem, where each colored line represents a different element $w_i^{(t)}$ or $|x_i^{(t)}|$ respectively. All details of the data generation process, as well as comprehensive attendant analyses, are deferred to the supplementary. To summarize here though, in the top plot the elements of $\boldsymbol{w}^{(t)}$, which represent the non-negative weights forming the outer-loop majorization step from (6) and reflect coarse correlation structure in $\boldsymbol{\Phi}$, converge very quickly ($\sim$3-5 iterations). Moreover, the observed bifurcation of magnitudes ultimately helps to screen many (but not necessarily all) elements of $\boldsymbol{x}^{(t)}$ that are the most likely to be zero in the maximally sparse representation (i.e., a stable, higher weighting value $w_i^{(t)}$ is likely to eventually cause $x_i^{(t)} \to 0$). In contrast, the actual coefficients $\boldsymbol{x}^{(t)}$ themselves converge much more slowly, with final destinations still unclear even after 50+ iterations. Hence $\boldsymbol{w}^{(t)}$ need not be continuously updated after rapid initial convergence, provided that we retain a memory of the optimal value during periods when it is static.

This discrepancy in convergence rates occurs in part because, as mentioned previously, the gate and cell updates do not fully solve the inner-loop weighted $\ell_1$ optimization needed to compute a globally optimal $\boldsymbol{x}^{(t+1)}$ give $\boldsymbol{w}^{(t)}$. Varying the number of inner-loop iterations, meaning additional executions

of $(8)-(11)$ with $\boldsymbol{w}^{(t)}$ fixed, is one heuristic for normalizing across different trajectory frequencies, but this requires additional computational overhead, and prior knowledge is needed to micro-manage iteration counts for either efficiency or final estimation quality. With respect to the latter, we conduct additional experiments in the supplementary which reveal that indeed the number of inner-loop updates per outer-loop cycle can affect the quality of sparse solutions, with no discernible rule of thumb for enhancing solution quality.[5] For example, navigating around suboptimal local minima could require adaptively adjusting the number inner-loop iterations in subtle, non-obvious ways. We therefore arrive at an unresolved state of affairs:

1. The latent variables which define SBL iterations can potentially follow optimization trajectories with radically different time scales, or both long- and short-term dependencies.

2. But there is no intrinsic mechanism within the SBL framework itself (or most multi-loop optimization problems in general either) for *automatically* calibrating the differing time scales for optimal performance.

These same issues are likely to arise in other non-convex multi-loop optimization algorithms as well. It therefore behooves us to consider a broader family of model structures that can adapt to these scales in a data-dependent fashion.

### 3.2 Modeling via Gated Feedback Nets

In addressing this fundamental problem, we make the following key observation: *If the trajectories of various latent variables can be interpreted as activations passing through an RNN with both long- and short-term dependencies, then in developing a pipeline for optimizing such trajectories it makes sense to consider learning deep architectures explicitly designed to adaptively model such characteristic sequences.* Interestingly, in the context of sequence prediction, the clockwork RNN (CW-RNN) has been proposed to cope with temporal dependencies engaged across multiple scales [25]. As shown in the supplementary however, the CW-RNN enforces dynamics synced to pre-determined clock rates exactly analogous to the fixed, manual schedule for terminating inner-loops in existing multi-loop iterative algorithms such as SBL. So we are back at our starting point.

Fortunately though, the gated feedback RNN (GF-RNN) [12] was recently developed to update the CW-RNN with an additional set of gated connections that, in effect, allow the network to learn its own clock rates. In brief, the GF-RNN involves stacked LSTM layers (or somewhat simpler gated recurrent unit (GRU) layers [11]), that are permitted to communicate bilaterally via additional, data-dependent gates that can open and close on different time-scales. In the context of SBL, this means that we no longer need strain a specialized LSTM structure with the burden of coordinating trajectory dynamics. Instead, we can stack layers that are, at least from a conceptual standpoint, designed to reflect the different dynamics of disparate variable sets such as $\boldsymbol{w}^{(t)}$ or $\boldsymbol{x}^{(t)}$. In doing so, we are then positioned to learn new SBL update rules from training pairs $\{\boldsymbol{y}, \boldsymbol{x}^*\}$ as described previously. At the very least, this structure should include SBL-like iterations within its capacity, but of course it is also free to explore something even better.

### 3.3 Network Design and Training Protocol

We stack two gated recurrent layers loosely designed to mimic the relatively fast SBL adaptation to basic correlation structure, as well as the slower resolution of final support patterns and coefficient estimates. These layers are formed from either LSTM or GRU base architectures. For the final output layer we adopt a multi-label classification loss for predicting $\text{supp}[\boldsymbol{x}^*]$, which is the well-known 'NP-hard' part of sparse estimation (determining final coefficient amplitudes just requires a simple least squares fit given the correct support pattern). Full network details are deferred to the supplementary, including special modifications to handle complex data as required by DOA applications.

For a given dictionary $\boldsymbol{\Phi}$ a separate network must be trained via SGD, to which we add a unique extra dimension of randomness via an online stochastic data-generation strategy. In particular, to create samples in each mini-batch, we first generate a vector $\boldsymbol{x}^*$ with random support pattern and nonzero amplitudes. We then compute $\boldsymbol{y} = \boldsymbol{\Phi}\boldsymbol{x}^* + \boldsymbol{\epsilon}$, where $\boldsymbol{\epsilon}$ is a small Gaussian noise component. This $\boldsymbol{y}$ forms a training input sample, while $\text{supp}[\boldsymbol{x}^*]$ represents the corresponding labels. For all

mini-batches, novel samples are drawn, which we have found boosts performance considerably over the fixed training sets used by current DNN approaches to sparse estimation (see supplementary).

# 4   Experiments

This section presents experiments involving synthetic data and two applications.

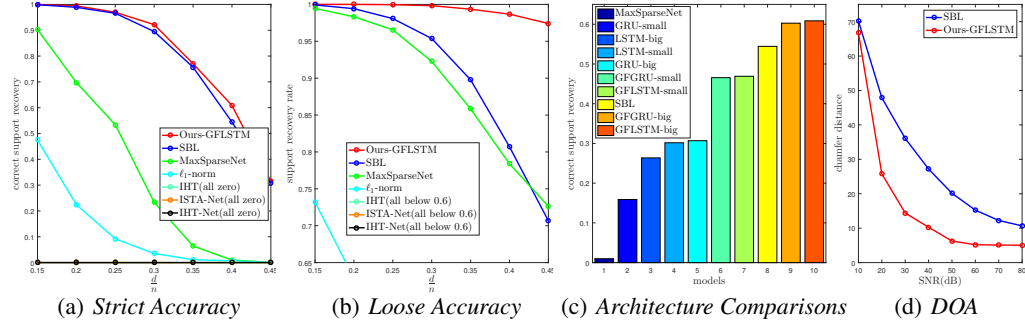

(a) *Strict Accuracy*    (b) *Loose Accuracy*    (c) *Architecture Comparisons*    (d) *DOA*

Figure 3: Plots (a), (b), and (c) show sparse recovery results involving synthetic correlated dictionaries. Plot (d) shows Chamfer distance-based errors [7] from the direction-of-arrival (DOA) experiment.

## 4.1   Evaluations via Synthetic Correlated Dictionaries

To reproduce experiments from [38], we generate correlated synthetic features via $\boldsymbol{\Phi} = \sum_{i=1}^{n} \frac{1}{i^2} \boldsymbol{u}_i \boldsymbol{v}_i^\top$, where $\boldsymbol{u}_i \in \mathbb{R}^n$ and $\boldsymbol{v}_i \in \mathbb{R}^m$ are drawn iid from a unit Gaussian distribution, and each column of $\boldsymbol{\Phi}$ is subsequently rescaled to unit $\ell_2$ norm. Ground truth samples $\boldsymbol{x}^*$ have $d$ nonzero elements drawn randomly from $\mathcal{U}[-0.5, 0.5]$ excluding the interval $[-0.1, 0.1]$. We use $n$=20, $m$=100, and vary $d$, with larger values producing a much harder combinatorial estimation problem (exhaustive search is not feasible here). All algorithms are presented with $\boldsymbol{y}$ and attempt to estimate supp$[\boldsymbol{x}^*]$. We evaluate using *strict accuracy*, meaning percentage of trials with exact support recovery, and *loose accuracy*, which quantifies the percentage of true positives among the top $n$ 'guesses' (i.e., largest predicted outputs).

Figures 3(a) and 3(b) evaluate our model, averaged across $10^5$ trials, against an array of optimization-based approaches: SBL [33], $\ell_1$ norm minimization [4], and IHT [5]; and existing learning-based DNN models: an ISTA-inspired network [20], an IHT-inspired network [34], and the best maximal sparsity net (MaxSparseNet) from [38] (detailed settings in the supplementary). With regard to strict accuracy, only SBL is somewhat competitive with our approach and other learning-based models are much worse; however, using loose accuracy our method is far superior than all others. Note that this is the first approach we are aware of in the literature that can convincingly outperform SBL recovering sparse solutions when a heavily correlated dictionary is present, and we hypothesize that this is largely possible because our design principles were directly inspired by SBL itself.

To isolate architectural factors affecting performance we conducted ablation studies: (i) with or without gated feedback, (iii) LSTM or GRU cells, and (iii) small or large ($4\times$) model size; for each model type, the small and respectively large versions have roughly the same number of parameters. The supplementary also contains a much broader set of self-comparison tests. Figure 3(c), which shows strict accuracy results with $d/n = 0.4$, indicates the importance of gated feedback and to a lesser degree network size, while LSTM and GRU cells perform similarly as expected.

## 4.2   Practical Application I: Direction-of-Arrival (DOA) Estimation

DOA estimation is a fundamental problem in sonar/radar processing [28]. Given an array of $n$ omnidirectional sensors with $d$ signal waves impinging upon them, the objective is to estimate the angular direction of the wave sources with respect to the sensors. For certain array geometries and known propagation mediums, estimation of these angles can be mapped directly to solving (2) in the complex domain. In this scenario, the $i$-th column of $\boldsymbol{\Phi}$ represents the sensor array output (a point in $\mathbb{C}^n$) from a hypothetical source with unit strength at angular location $\theta_i$, and can be computed using wave progagation formula [28]. The entire dictionary can be constructed by concatenating columns associated with angles forming some spacing of interest, e.g., every $1°$ across a half circle, and will be highly correlated. Given measurements $\boldsymbol{y} \in \mathbb{C}^n$, we can solve (2), with $\lambda$ reflecting the noise level.

The indexes of nonzero elements of $\boldsymbol{x}^*$ will then reveal the angular locations/directions of putative sources.

Recently SBL-based algorithms have produced state-of-the-art results solving the DOA problem [14, 19, 39], and we compare our approach against SBL here. We apply a typical experimental design from the literature involving a uniform linear array with $n = 10$ sensors; see supplementary for background and details on how to compute $\boldsymbol{\Phi}$, as well as specifics on how to adapt and train our GFLSTM using complex data. Four sources are then placed in random angular locations, with nonzero coefficients at $\{\pm 1 \pm i\}$, and we compute measurements $\boldsymbol{y} = \boldsymbol{\Phi}\boldsymbol{x}^* + \boldsymbol{\epsilon}$, with $\boldsymbol{\epsilon}$ chosen from a complex Gaussian distribution to produce different SNR. Because the nonzero positions in $\boldsymbol{x}^*$ now have physical meaning, we apply the Chamfer distance [7] as the error metric, which quantifies how close we are to true source locations (lower is better). Figure 3(d) displays the results, where our learned network outperforms SBL across a range of SNR values.

Table 1: Photometric stereo results

| Algorithm | Average angular error (degrees) | | | | | | Runtime (sec.) | | | | | |
|---|---|---|---|---|---|---|---|---|---|---|---|---|
| | Bunny | | | Caesar | | | Bunny | | | Caesar | | |
| | r=10 | r=20 | r=40 | r=10 | r=20 | r=40 | r=10 | r=20 | r=40 | r=10 | r=20 | r=40 |
| SBL | 4.02 | 1.86 | **0.50** | 4.79 | 2.07 | **0.34** | 35.46 | 22.66 | 32.20 | 86.96 | 64.67 | 90.48 |
| MaxSparseNet | 1.48 | 1.95 | 1.20 | 3.51 | 2.51 | 1.18 | 0.90 | 0.87 | 0.92 | 2.13 | 2.12 | 2.20 |
| Ours | **1.35** | **1.55** | 1.12 | **2.39** | **1.80** | 0.60 | **0.63** | **0.67** | **0.85** | **1.48** | **1.70** | **2.08** |

## 4.3 Practical Application II: 3D Geometry Recovery via Photometric Stereo

Photometric stereo represents another application domain whereby approximately solving (2) using SBL has recently produced state-of-the-art results [24]. The objective here is to recover the 3D surface normals of a given scene using $r$ images taken from a single camera but with different lighting conditions. Under the assumption that these images can be approximately decomposed into a diffuse Lambertian component and sparse corruptions such as shadows and specular highlights, then surface normals at each pixel can be recovered using (2) to isolate these sparse factors followed by a final least squares post-processing step [24]. In this context, $\boldsymbol{\Phi}$ is constructed using the known camera and lighting geometry, and $\boldsymbol{y}$ represents intensity measurements for a given pixel across images projected onto the nullspace of a special transposed lighting matrix (see supplementary for more details and our full experimental design). However, because a sparse regression problem must be computed for every pixel to recovery the full scene geometry, a fast, efficient solver is paramount.

We compare our GFLSTM model against both SBL and the MaxSparseNet [38] (both of which outperform other existing methods). Tests are performed using the 32-bit HDR gray-scale images of objects 'Bunny' ($256 \times 256$) and 'Caesar' ($300 \times 400$) as in [24]. For (very) weakly-supervised training data, we apply the same approach as before, only we use nonzero magnitudes drawn from a Gaussian, with mean and variance loosely tuned to the photometric stereo data, consistent with [38]. Results are shown in Table 1, where we observe in all cases the DNN models are faster by a wide margin, and in the hard cases cases (smaller $r$) our approach produces the lowest angular error. The only exception is with $r = 40$; however, this is a quite easy scenario with so many images such that SBL can readily find a near optimal solution, albeit at a high computational cost. See supplementary for error surface visualizations.

## 5 Conclusion

In this paper we have examined the structural similarities between multi-loop iterative algorithms and multi-scale sequence prediction neural networks. This association is suggestive of a learning process for a richer class of algorithms that employ multiple loops and latent states, such as the EM algorithm or general majorization-minimization approaches. For example, in a narrower sense, we have demonstrated that specialized gated recurrent nets carefully patterned to reflect the multi-scale optimization trajectories of multi-loop SBL iterations can lead to a considerable boost in both accuracy and efficiency. Note that simpler first-order, gradient descent-style algorithms can be ineffective when applied to sparsity-promoting energy functions with a combinatorial number of bad local optima and highly concave or non-differentiable surfaces in the neighborhood of minima. Moreover, implementing smoother approximations such as SBL with gradient descent is impractical since each gradient calculation would be prohibitively expensive. Therefore, recent learning-to-learn approaches such as [1] that explicitly rely on gradient calculations are difficult to apply in the present setting.

**Acknowledgments**

This work was accomplished while Hao He was an intern at Microsoft Research, Beijing.

## Footnotes

[1]Or global convergence to some stationary point with mild additional assumptions [31].

[2]If we allow for peephole connections [18], it is possible to reverse these roles; however, for simplicity and the most direct mapping to LSTM cells we do not pursue this alternative here.

[3]Incidently, this association also suggests that the role of hidden cell updates in LSTM networks can be reinterpreted as an analog to the expectation step (or E-step) for estimating hidden data in a suitably structured EM algorithm.

[4]For $\boldsymbol{w}^{(t)}$ the result of Proposition 1 suggests that these weights can be computed as the solution of a simple regularized regression problem, which can easily be replaced with a small network analogous to that used in [18]; similarly for $\boldsymbol{\nu}^{(t)}$.

[5]In brief, these experiments demonstrate a situation where executing either 1, 10, or 1000 inner-loop iterations per outer loop fails to produce the optimal solution, while 100 inner-loop iterations is successful.

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
