[Supplementary Material]

# From Bayesian Sparsity to Gated Recurrent Nets
## −Supplementary File−

**Hao He**                                          HAOHE@MIT.EDU
*Massachusetts Institute of Technology*

**Bo Xin**                                          JIMXINBO@GMAIL.COM
*Microsoft Research, Beijing*

**Satoshi Ikehata**                         SATOSHI.IKEHATA@GMAIL.COM
*National Institute of Informatics, Tokyo*

**David Wipf**                                      DAVIDWIPF@GMAIL.COM
*Microsoft Research, Beijing*

## 1. Contents

This document includes the following supporting materials, which we hope to later aggregate into a self-contained journal version:

- **Section 2 - Notation**

- **Section 3 - The Dynamics of SBL Iterations:** This includes the quantification of trajectory time-scale differences, adaptations to correlated dictionaries, and a demonstration of the potential value of data-dependent schedules for coordinating inner- and outer-loops.

- **Section 4 - Clockwork Networks and Fixed Inner-Loop Iterations:** We describe the intimate relationship between clockwork recurrent neural networks (RNN) and the typical manual tuning of inner-loop iterations during classical optimization.

- **Section 5 - Modeling and Training Details**

- **Section 6 - Experimental Details for Direction-of-Arrival (DOA) Estimation**

- **Section 7 - Experimental Details for 3D Geometry Recovery via Photometric Stereo**

- **Section 8 - Additional Experiments and Self-Comparisons**

- **Section 9 - Technical Proofs**

## 2. Notation

All equation numbers referencing back to the main paper will be prefixed with an 'M' to avoid confusion, i.e, (M.#) will refer to equation (#) from the main text. Similar notation differentiates sections, tables, and figures, e.g., Section M.#, etc.

## 3. The Dynamics of SBL Iterations

This section empirically examines the trajectories of SBL iterations produced via the rules derived in Section M.2.3.

### 3.1 Large Timescale Differences

Here we present a synthetic experiment that highlights the different time scales upon which SBL latent variables may fluctuate over the course of a typical optimization trajectory. The experimental design is as follows: First we generate a dictionary $\boldsymbol{\Phi}$ via

$$\boldsymbol{\Phi} = \widetilde{\boldsymbol{\Phi}} \boldsymbol{B} \boldsymbol{D}, \tag{1}$$

where $\widetilde{\boldsymbol{\Phi}} \in \mathbb{R}^{50 \times 100}$ has iid elements drawn from $\mathcal{N}(0,1)$; $\boldsymbol{B} \in \mathbb{R}^{100 \times 100}$ is a block-diagonal matrix with 20, $5 \times 5$ blocks, each with unit diagonals and off-diagonals set to 0.9; and $\boldsymbol{D} \in \mathbb{R}^{100 \times 100}$ is a fully diagonal matrix that re-scales each column of the final $\boldsymbol{\Phi}$ to have unit $\ell_2$ norm, and finally multiplies by a random sign pattern. This process ensures that $\boldsymbol{\Phi}$ will encompass 20 clusters of 5 adjacent columns each, with strong correlations introduced via $\boldsymbol{B}$. We then generate a sparse random vector $\boldsymbol{x}^* \in \mathbb{R}^{100}$ such that $\|\boldsymbol{x}^*\|_0 = 10$, where the nonzero positions are randomly aligned with 10 different clusters, and the nonzero values have unit magnitude. We next compute $\boldsymbol{y} = \boldsymbol{\Phi} \boldsymbol{x}^*$ and apply the revised SBL iterations from Section M.2.3 with $\lambda = 0.01$ (or a rather arbitrary small value), $\boldsymbol{\alpha}(\boldsymbol{\gamma}) = \mathbf{1}$, and $\boldsymbol{\beta}(\boldsymbol{\gamma}) = \mathbf{0}$.

Figure 1 displays the trajectories of both $\boldsymbol{w}^{(t)}$ (left subplot) and $\boldsymbol{x}^{(t)}$ (right subplot) for $t = 1, \ldots, 100$ during execution of (M.8)$-$(M.11) in the main text. As mentioned previously, the weights $\boldsymbol{w}^{(t)}$ serve to incrementally focus a sequence of $\ell_1$ minimization problems towards likely nonzero elements of $\boldsymbol{x}^*$ via the process defined by (M.7). Unlike other existing iterative reweighted $\ell_1$ approaches, with SBL these weights quickly (within just a few iterations) partition into two groups, one with smaller values near 1.0, the other with larger values in the 8-10 range (see left subplot).

Moreover, upon closer examination we found that the index $i$ of all weights $w_i^{(t)}$ with a value near 1.0 correspond with dictionary columns $\boldsymbol{\phi}_i$ in a cluster where some $x_j^* \neq 0$. In contrast, all weights with a large value are associated with dictionary columns in clusters where all $x_j^* = 0$. Consequently, these weights reflect in some sense the correct support at the cluster level, and introduce a more severe penalty to coefficients associated with what should ideally be inactive clusters. This then allows subsequent $\ell_1$ iterations to more narrowly learn the correct support pattern *within* these favored clusters, providing empirical support to the arguments made in Section M.2.2 for the efficacy of SBL in dealing with correlated dictionary structure.

However, the secondary learning of final coefficient values within the correct clusters occurs at a radically different time scale as shown in the right subplot of Figure 1. Here we observe that even after 50 iterations it is still not clear to which final value each coefficient

Figure 1: Illustration of the different time scales upon which SBL latent variables operate under a multi-resolution, clustered dictionary model. *Left*: Trajectories of the weights $\boldsymbol{w}^{(t)}$ across 100 iterations (each colored line represents a different element $w_i^{(t)}$). Within just a few iterations, these values have completely converged and accurately reflect the true cluster-level support pattern, namely, high values in the 8-10 range represent weights associated with false clusters (hence a high penalty/weight), while low values around 1.0 indicate weights associated with correct clusters. *Right*: Corresponding trajectories of $\left|\boldsymbol{x}^{(t)}\right|$ (a different colored line indicates a different element $|x_i^{(t)}|$). Here we observe that even after 50 iterations, it is not entirely clear to what value each element will finally converge to. From these plots it is readily apparent that after 5 iterations, it is no longer necessary to update $\boldsymbol{w}^{(t)}$, provided the network is capable of memorizing the stored value from prior iterations, while $\boldsymbol{x}^{(t)}$ must be updated even beyond 100 iterations for full convergence.

magnitude $x_i^{(t)}$ will converge too, for example, 0.0 or 1.0. Therefore we may conclude that, although the weights $\boldsymbol{w}^{(t)}$ may rather quickly proceed to values that reflect the correlation structure of the dictionary, the final coefficient estimates take much longer to resolve. Moreover, during this time, to be effective the iterations must 'remember' the correct value of $\boldsymbol{w}^{(t)}$, even if continued updates are not necessary after rapid initial convergence.

### 3.2 The Potential Value of an Adaptive Updating Schedule

Although the previous experiment served to expose the differing scales of subsets of latent variables, it did not provide any indication of how these different scales may actually impact final estimation accuracy. For example, suppose we were able to speed up the convergence of $\boldsymbol{x}$ for any fixed value of $\boldsymbol{w}$, would this improve the overall performance? The present simulation directly addresses this issue.

We begin with a similar experimental design as used in Section 3.1, although we reduce the dimensions for visualization purposes. In brief, we choose $\boldsymbol{\Phi} \in \mathbb{R}^{10 \times 20}$, formed from 10 clusters of size 2 each, and $\|\boldsymbol{x}^*\|_0 = 3$. Figure 2(a) displays the optimal support pattern, whereby a '1' indicates the location of a true nonzero, and a zero otherwise. Without loss of generality, we have also reordered the dictionary columns such that the first three columns serve as nonzero locations.

Figure 2: Estimated support patterns obtained under varying numbers of inner-loop iterations $K$. A blue bar indicates that the corresponding index is associated with a nonzero element of $\hat{x}$. We observe that when $K \in \{1, 10, 1000\}$, the estimated support pattern is incorrect; see plots (b), (c), and (e). In contrast, $K = 100$ produces the correct result; see plot (d). Hence adaptively terminating these inner-loop iterations in a data-dependent fashion can potentially improve the final result.

We display recovery results using a modified version of the SBL implementation from Section M.2.3, whereby the gate and cell update steps, which are associated with the weighted $\ell_1$ norm problem from (M.6), are applied in a varying number $K$ of inner-loop iterations. More specifically, with $\boldsymbol{\alpha}(\boldsymbol{\gamma}) = \mathbf{1}$, and $\boldsymbol{\beta}(\boldsymbol{\gamma}) = \mathbf{0}$ and $\boldsymbol{w}^{(t)}$ fixed, these additional, reduced inner-loop iterations consist of simply computing

$$
\begin{aligned}
\boldsymbol{\nu}^{(t,k)} &\leftarrow \boldsymbol{x}^{(t+1,k)} + \mu \boldsymbol{\Phi}^\top \left( \boldsymbol{y} - \boldsymbol{\Phi} \boldsymbol{x}^{(t+1,k)} \right) \\
\boldsymbol{\sigma}_{in}^{(t,k)} &\leftarrow \left[ \left| \boldsymbol{\nu}^{(t,k)} \right| - 2\lambda \boldsymbol{w}^{(t)} \right]_+ \\
\bar{\boldsymbol{x}}^{(t+1,k)} &\leftarrow \text{sign} \left[ \boldsymbol{\nu}^{(t,k)} \right] \\
\boldsymbol{x}^{(t+1,k+1)} &\leftarrow \boldsymbol{\sigma}_{in}^{(t,k)} \odot \bar{\boldsymbol{x}}^{(t+1,k)},
\end{aligned}
\tag{2}
$$

from $k = 1, \ldots, K$, where $\boldsymbol{x}^{(t+1,1)} \triangleq \boldsymbol{u}^{(t)}$. For any fixed $\boldsymbol{w}^{(t)}$, these iterations are guaranteed to converge to a minimum of (M.6), and in light of the experiment from Section 3.1, can be viewed as a direct way of constricting or shrinking the x-axis of Figure 1(right). Indeed, for sufficiently large $K$, once $\boldsymbol{w}$ has converged, $\boldsymbol{x}$ will immediately follow. But is this necessarily a desirable course of action?

Suplots (b)−(e) of Figure 2 show the support patterns of the $\hat{x}$ estimate obtained via this procedure using $K \in \{1, 10, 100, 1000\}$. Only the $K = 100$ case produces a perfect recovery with matching support. It therefore follows that inner-loop iterations, when interpreted as a tunable sequence, have the potential to improve performance. Of course in advance we have no way of knowing what the best $K$ might be. But at least we do know that the $K = 1$ case which emerges from the original LSTM template need not be optimal.

## 4. Clockwork Networks and Fixed Inner-Loop Iterations

In the context of sequence prediction, the *clockwork recurrent neural network* (CW-RNN) has been proposed to cope with temporal dependencies engaged across multiple scales [8]. In its most basic form, the CW-RNN begins with input, hidden, and output layers which,

just like a regular RNN, are defined by

$$\boldsymbol{h}^{(t+1)} \;=\; f_H\left(\boldsymbol{W}_H \cdot \boldsymbol{h}^{(t)} + \boldsymbol{W}_I \cdot \boldsymbol{z}^{(t)}\right) \tag{3}$$

$$\boldsymbol{v}^{(t+1)} \;=\; f_O\left(\boldsymbol{W}_O \cdot \boldsymbol{h}^{(t+1)}\right), \tag{4}$$

where $\boldsymbol{z}^{(t)}$ is an input vector at time $t$, $\boldsymbol{h}^{(t)}$ represents hidden layer activations, $\boldsymbol{v}^{(t+1)}$ the output, and $\{\boldsymbol{W}_I, \boldsymbol{W}_H, \boldsymbol{W}_O\}$ are input, hidden, and output weight matrices respectively. Likewise, $f_H$ and $f_O$ are the corresponding nonlinear activation functions. What differentiates the CW-RNN from this vanilla structure, is that $\boldsymbol{W}_I$ and $\boldsymbol{W}_H$ are each partitioned into $g$ different temporarlly-varying block-rows[1] as

$$\boldsymbol{W}_I = \begin{bmatrix} \boldsymbol{W}_{I_1}^{(t)} \\ \vdots \\ \boldsymbol{W}_{I_g}^{(t)} \end{bmatrix}, \quad \boldsymbol{W}_H = \begin{bmatrix} \boldsymbol{W}_{H_1}^{(t)} \\ \vdots \\ \boldsymbol{W}_{H_g}^{(t)} \end{bmatrix}, \tag{5}$$

which naturally defines a corresponding segmentation of the hidden variables as

$$\boldsymbol{h}^{(t)} = \begin{bmatrix} \boldsymbol{h}_1^{(t)} \\ \vdots \\ \boldsymbol{h}_g^{(t)} \end{bmatrix} \tag{6}$$

such that, assuming separable nonlinearities,

$$\boldsymbol{h}_i^{(t+1)} = f_H\left(\boldsymbol{W}_{H_i}^{(t)} \cdot \boldsymbol{h}^{(t)} + \boldsymbol{W}_{I_i}^{(t)} \cdot \boldsymbol{z}^{(t)}\right), \; \forall i = 1, \ldots, g. \tag{7}$$

Additionally, each block is assigned and 'update period' $T_i$ that governs the structure across each time step $t$ via

$$\boldsymbol{W}_{I_i}^{(t)} \;=\; \begin{cases} \widetilde{\boldsymbol{W}}_{I_i} & \text{for } (t \bmod T_i) = 0 \\ [\boldsymbol{0}_1, \ldots, \boldsymbol{0}_g] & \text{otherwise} \end{cases} \tag{8}$$

and

$$\boldsymbol{W}_{H_i}^{(t)} \;=\; \begin{cases} \widetilde{\boldsymbol{W}}_{H_i} & \text{for } (t \bmod T_i) = 0 \\ [\boldsymbol{0}_1, \ldots, \boldsymbol{0}_{i-1}, \boldsymbol{I}, \boldsymbol{0}_{i+1}, \ldots, \boldsymbol{0}_g] & \text{otherwise}. \end{cases} \tag{9}$$

In brief, these weight expressions ensure that for all $i$ we have

$$\boldsymbol{h}_i^{(t+1)} \;=\; \begin{cases} f_H\left(\widetilde{\boldsymbol{W}}_{H_i} \cdot \boldsymbol{h}^{(t)} + \widetilde{\boldsymbol{W}}_{I_i} \cdot \boldsymbol{z}^{(t)}\right) & \text{for } (t \bmod T_i) = 0 \\ \boldsymbol{h}_i^{(t)} & \text{otherwise}. \end{cases} \tag{10}$$

This formulation allows the CW-RNN to handle different temporal features by assigned different $T_i$ to different blocks. For example, a block designed to model high-frequency

---

1. A column-wise block structure may also be assumed if desired; however, this is not required for what follows herein.

dynamics may assume $T_i = 1$, while slowly-varying components can be captured using $T_i \gg 1$. The latter implies that for most iterations, the block hidden state $\boldsymbol{h}_i^{(t)}$ is not updated allowing for hard-coded long-term memory of such low-frequency dynamics.

This prescription exactly reflects the basic anatomy of an algorithm with $g$ nested loops, each loop being characterized by its own set of latent variables $\boldsymbol{h}_i^{(t)}$. As a simple example, consider the SBL updates equipped with an inner-loop as in Section 3.2. If we define $\boldsymbol{h}_1^{(t)} = \boldsymbol{x}^{(t)}$, $\boldsymbol{h}_2^{(t)} = \boldsymbol{w}^{(t)}$, $\boldsymbol{z}^{(t)} = \boldsymbol{y}$, adopt $T_1 = 1$ and $T_2 = K$, and relabel the iteration numbers via a single consistent index (i.e., we collapse $k$ and $t$ into a single index), then $\boldsymbol{w}^{(t)}$ will only be updated once every $T_2$ time-steps, while $\boldsymbol{x}^{(t)}$ will be updated at all $t$, and the basic scheduling is identical. The only difference is that the layer-wise filters and nonlinearities are somewhat more specialized for the SBL context.

## 5. Modeling and Training Details

In this section we first describe the basic gated feedback RNN structure, followed by our particular model architecture including extensions to handle complex data. We conclude with training details and experimental settings.

### 5.1 Gated Feedback RNN Structure

The gated feedback RNN cell [3] is a key component of our model. Detailed computing flows for a gated feedback LSTM cell (GFLSTM), which represents one particular specialization that is used in all our experiments, follow as

$$
\begin{aligned}
\boldsymbol{c}_j^{(t)} &= \boldsymbol{f}_j^{(t)} \odot \boldsymbol{c}_j^{(t-1)} + \boldsymbol{i}_j^{(t)} \odot \tilde{\boldsymbol{c}}_j^{(t)} \\
\boldsymbol{h}_j^{(t)} &= \boldsymbol{o}_j^{(t)} \odot \mathrm{Tanh}(\boldsymbol{c}_j^{(t)}) \\
\boldsymbol{i}_j^{(t)} &= \sigma(\boldsymbol{W}_{ij}\boldsymbol{a}_j^{(t)} + \boldsymbol{U}_{ij}\boldsymbol{h}_j^{(t-1)}) \\
\boldsymbol{f}_j^{(t)} &= \sigma(\boldsymbol{W}_{fj}\boldsymbol{a}_j^{(t)} + \boldsymbol{U}_{fj}\boldsymbol{h}_j^{(t-1)}) \\
\boldsymbol{o}_j^{(t)} &= \sigma(\boldsymbol{W}_{oj}\boldsymbol{a}_j^{(t)} + \boldsymbol{U}_{oj}\boldsymbol{h}_j^{(t-1)}) \\
\boldsymbol{g}_{i \to j}^{(t)} &= \sigma(\boldsymbol{W}_{g_j}\boldsymbol{a}_j^{(t)} + \boldsymbol{U}_{g_{i \to j}}\boldsymbol{H}^{(t-1)}) \\
\tilde{\boldsymbol{c}}_j^{(t)} &= \mathrm{Tanh}(\boldsymbol{W}_{cj-1 \to j}\boldsymbol{h}_{j-1}^{(t)} + \sum_{i=1}^{r} \boldsymbol{g}_{i \to j}^{(t)} \odot \boldsymbol{U}_{ci \to j}\boldsymbol{h}_i^{(t-1)}),
\end{aligned}
\tag{11}
$$

where $r$ is the number of stacked LSTM cells, subscript $j$ is the LSTM cell index in the stack, while superscript $(t)$ indicates the time point. Therefore $\boldsymbol{h}_j^{(t)}$ and $\boldsymbol{c}_j^{(t)}$ denote the *hidden state* and *memory cell* of $j$-th LSTM unit in the stack at time $t$. And we denote $\boldsymbol{a}_j^{(t)}$ as the input of the $j$-th LSTM cell, such that $\boldsymbol{a}_j^{(t)} = \boldsymbol{h}_{j-1}^{(t)}$ ($\forall j > 1$) and $\boldsymbol{a}_1^{(t)} = \boldsymbol{y}$. Besides conventional designs like an *input gate* $\boldsymbol{i}_j^{(t)}$, *forget gate* $\boldsymbol{f}_j^{(t)}$, and *output gate*, $\boldsymbol{o}_j^{(t)}$, the stack of GFLSTM cell also includes an extra *global gate* computed from input $\boldsymbol{a}_j^{(t)}$ and $\boldsymbol{H}^{(t-1)} = [\boldsymbol{h}_1^{(t-1)}, .., \boldsymbol{h}_r^{(t-1)}]$, the concatenation of all the hidden states from the previous time step $t-1$. Each $\boldsymbol{g}_{i \to j}^{(t)}$ controls the flow from $\boldsymbol{h}_i^{(t-1)}$ to $\boldsymbol{h}_j^{(t)}$, that is, the cross-layer

feedback. To make it concise, we can denote the whole computing flow of these $r$ LSTM cells using the function $f_{GFLSTM}$ as

$$
\begin{aligned}
f_{GFLSTM}(\boldsymbol{H}^{(t-1)}, \boldsymbol{y}; \boldsymbol{\theta}_{GFLSTM}) &= [\boldsymbol{q}^{(t)}, \boldsymbol{H}^{(t)}] \\
\boldsymbol{\theta}_{GFLSTM} &= [\boldsymbol{W}_{ij}, \boldsymbol{W}_{fj}, \boldsymbol{W}_{oj}, \boldsymbol{U}_{ij}, \boldsymbol{U}_{fj}, \boldsymbol{U}_{oj}, \boldsymbol{W}_{gj}, \boldsymbol{U}_{g_{i \to j}}, \boldsymbol{W}_{cj-1 \to j}, \boldsymbol{U}_{ci \to j}] \\
\boldsymbol{q}^{(t)} &= \boldsymbol{h}_r^{(t)}.
\end{aligned}
\tag{12}
$$

## 5.2 Proposed Model Architecture and Extensions

**Basic Model:** Although our model consists of RNN cells, once we fix the number of unfolding steps, it essentially becomes a feed-forward network. As shown in Figure 3, during the forward stage, the input is broadcast to the lowest RNN cell at each unrolled step. After the model generates its outputs at each unrolled step, they will be concatenated and fed into a *fully connected* layer to produce the final prediction. Since we opt to predict the support pattern $\text{supp}[\boldsymbol{x}^*] = \{i : x_i^* \neq 0\}$, we view the problem as an multi-label classification task and append a *softmax* layer on top of the fully connected layer. We formalize this process as

$$
\begin{aligned}
[\boldsymbol{q}^{(t)}, \boldsymbol{H}^{(t)}] &= f_{rnn}(\boldsymbol{H}^{(t-1)}, \boldsymbol{y}; \boldsymbol{\theta}_{rnn}) \\
\boldsymbol{p} &= f_{pred}([\boldsymbol{q}^{(1)}, \boldsymbol{q}^{(2)}, .., \boldsymbol{q}^{(T)}]; \boldsymbol{\theta}_{pred}) \\
f_{pred}(\boldsymbol{q}^{(all)}, \boldsymbol{\theta}_{pred}) &= softmax(\boldsymbol{W}_{pred}\boldsymbol{q}^{(all)} + \boldsymbol{b}_{pred}) \\
\boldsymbol{q}^{(all)} &= [\boldsymbol{q}^{(1)}, \boldsymbol{q}^{(2)}, .., \boldsymbol{q}^{(T)}],
\end{aligned}
\tag{13}
$$

where $\boldsymbol{\theta}_{rnn}, \boldsymbol{\theta}_{pred} = [\boldsymbol{W}_{pred}, \boldsymbol{b}_{pred}]$ are the parameters of the RNN units and the fully connected layer respectively. $\boldsymbol{q}^{(t)}, \boldsymbol{h}^{(t)}$ denote the output of the RNN units and hidden state at each time step $t$. In practice, we simply take the RNN's top layer hidden states as its output $\boldsymbol{q}^{(t)}$. $f_{rnn}$ represents the forward process of the RNN, which is defined by the exact structure of the RNN-cell.

Complex Value Extension: In many real applications of sparse recovery, the format of the inputs may vary. For example, the inputs to the DOA problem of interest are complex numbers. We propose to deal with complex-value inputs by what we call *model complexification*. Specifically, RNN units consist of matrix multiplication and non-linear activations, both of which have their complex value counterparts. Thus we propose to use complex-value operations in the RNN units before finally concatenating the real and imaginary part of the outputs as the feature for the final prediction. This method is inspired by SBL, which handles real and complex value inputs with the same operator. We argue that this method is better than simply concatenating the real and imaginary parts of the input and using a regular real-valued RNN (of double the size) for prediction, since in this way the links between real and imaginary parts of a complex number are broken and therefore the RNN may potentially have to learn these links by itself, which can be viewed as an unnecessary distraction.

Figure 3: Proposed model architecture using gated feedback LSTM cells

## 5.3 Training Details

We apply a unified training framework for all different approaches. In our experiments, models are implemented using Torch7 and experiments are run on a single NVIDIA Tesla K40M GPU card.

**Training Hyperparameters:** To provide consistency with the concept of epoch from [14], our models are trained via $600000/250 = 2400$ batches with batch size equal to 250. Typically, with 400 epochs (or 800 epochs in some extreme cases) of RMSprop optimization, we converge to a satisfactory performance level, with a default initial learning rate of 0.002, factored by 0.25 every 50 epochs after the first 250 epochs of training.

**Model Hyperparameters:** As for model architecture, there exists the following hyperparameters: the number of RNN hidden units $h$, the number of stacked RNN layers $r$, and the number of RNN-cell unfolding steps $T$. In most of our experiments, we control model capacity mainly by the size of hidden states with a fixed number of layers $r = 2$ and unfolding steps $T = 11$. In section 8.2 though, we provide more detailed ablation studies on how the number of RNN layers, unrolling steps and hidden units affects the performance.

**A Useful Training Heuristic:** When training with a fixed-sized dataset, as existing learning approaches to sparse estimation do [4, 10, 14], there is always the risk of overfitting. The gap between the error on training and validation sets with a fixed dataset is shown via the blue curves in Figure 4(a)) on a representative learning problem. However, since we are free to generate online as much training data as we want in the sparse estimation context (and other related problems), at every epoch we can always use a new, unseen batch. This simple strategy completely closes the gap (the red curves) with negligible computational overhead. Figure 4(b) displays the resulting improvement on performance, as measured by the percentage of trials whereby the entire support pattern is correctly estimated (i.e. *strict accuracy*).

(a)　　　　　　　　　　　(b)

Figure 4: Demonstration of online training heuristic benefits

## 6. Experimental Details for Direction-of-Arrival (DOA) Estimation

This section contains DOA background information, followed by experimental design and training details related to this application.

### 6.1 Background

Direction-of-arrival (DOA) estimation for sonar and radar application can be formulated by the observation model

$$\boldsymbol{y}(t) = \sum_{k=1}^{d} s_k(t) f(\theta_k^*) + \boldsymbol{\epsilon}(t), \tag{14}$$

where $\boldsymbol{y}(t) \in \mathbb{C}^n$ is the measured sonor/radar signal at time $t$, $f : \mathbb{R} \to \mathbb{C}^n$, and $d$ is number of source waveforms whose magnitudes are $\boldsymbol{s}(t) = [s_1(t), ..., s_d(t)]^\top \in \mathbb{C}^d$ and angular locations are $\boldsymbol{\theta}^* = [\theta_1^*, ..\theta_d^*]^\top$ [9]. Although the location space $\Theta$ might be continuous, we may approximate it by a fixed sampling grid $\boldsymbol{\theta} = [\theta_1, .., \theta_m]$. Then the problem can be rewritten as the alternative observation model

$$\boldsymbol{y}(t) = \sum_{i=1}^{m} x_i(t) \boldsymbol{\phi}_i + \boldsymbol{\epsilon}(t) = \boldsymbol{\Phi} \boldsymbol{x}(t) + \boldsymbol{\epsilon}(t), \tag{15}$$

where $\boldsymbol{\Phi} = [\boldsymbol{\phi}_1, .., \boldsymbol{\phi}_m]$, $\boldsymbol{\phi}_i \triangleq f(\theta_i)$, and $\boldsymbol{x}(t) = [x_1(t), ..., x_m(t)]^\top$. With sufficient resolution provided, we assume every $\theta_k^*$ is contained in $\boldsymbol{\theta}$ such that $\boldsymbol{s}(t)$ becomes a collection of $d$ non-zero entries in $\boldsymbol{x}(t)$. Finally, the DOA estimation problem boils down to solving $\min_{\boldsymbol{x}} \|\boldsymbol{y} - \boldsymbol{\Phi}\boldsymbol{x}\|_2^2 + \lambda \|\boldsymbol{x}\|_0$, whereby nonzero elements in $\boldsymbol{x}^*$ will (approximately) correspond with locations $i$ whereby $\theta_i \approx \theta_k^*$ for some $k$.

### 6.2 Experimental Design

We make some natural assumptions in our experiment. First we consider the narrowband, far-field case which implies that incoming waves are approximately planar and each source

emanates from a single point. Furthermore, we assume our sensors are arranged having a linear, uniformly spaced array geometry, i.e., *uniform linear array*(ULA), and a known propagation medium. Then the measurement vector $\boldsymbol{y}(t)$ obtained by the sensors at time $t$ is given by (14), where the non-linear function $f$ is

$$f(\theta) = \left[e^{i\omega_0 \Delta_1(\theta)}, .., e^{i\omega_0 \Delta_n(\theta)}\right]^\top \quad \text{with}$$

$$\omega_0 \Delta_j(\theta) = 2\pi(j-1)\frac{Dcos(\theta)}{\lambda_0}, \forall j = 1, .., n, \tag{16}$$

and $\omega_0$, $\lambda_0$ are the central temporal frequency and the wavelength of signals respectively. Also, $\Delta_j(\theta)$ is the array-geometry-dependent time delay between the first sensor and the $j$-th sensor for a given angle $\theta \in [0, \pi]$, while $D$ is the distance between two nearby sensors in the ULA.

**Settings:** In our experiments, we set $m = 180$ allowing an angular resolution of $1°$ over the half circle, and $n = 10$ sensors with $D = 0.5\lambda_0$. The dictionary $\boldsymbol{\Phi}$ is constructed via (14) and (16) such that the $i$-th column represents the sensor array output from a hypothetical source of unit strength at angular location $\theta_i$. The number of different sources $d$ is set to 4, which represents a quite challenging problem with only 10 sensors; most sparse estimation algorithms will fail in this regime. Then we randomly pick four different source directions with magnitudes $\{\pm 1 \pm i\}$. Finally, a measurement vector $\boldsymbol{y} = \boldsymbol{\Phi x} + \boldsymbol{\epsilon}$ is calculated with complex Gaussian noise added to maintain a given signal noise ratio (SNR).

**Metric:** We apply the symmetric Chamfer distance [1] to evaluate the estimation quality with respect to the ground truth source directions. This distance between ground truth $\boldsymbol{\theta}^* = \{\theta_1^*, ..\theta_d^*\}$ and predictions $\hat{\boldsymbol{\theta}} = \{\hat{\theta}_1, ..\hat{\theta}_d\}$ is given by

$$dist(\boldsymbol{\theta}^*, \hat{\boldsymbol{\theta}}) = \sum_{\theta_1 \in \boldsymbol{\theta}^*} \min_{\theta_2 \in \hat{\boldsymbol{\theta}}} |\theta_1 - \theta_2| + \sum_{\theta_2 \in \hat{\boldsymbol{\theta}}} \min_{\theta_1 \in \boldsymbol{\theta}^*} |\theta_1 - \theta_2|. \tag{17}$$

**Training Details:** For DOA experiments, our model has LSTM cells with 200 hidden units and is trained 400 epoches following our default settings. For training data generation, we tried using noise levels in the intervals $[15dB, 30dB]$, $[20dB, 40dB]$, $[30dB, 60dB]$, and $[60dB, 80dB]$, and then chose the best results.

## 7. Experimental Details for 3D Geometry Recovery via Photometric Stereo

This section describes our photometric stereo experiments in more detail, including error surface visualizations.

### 7.1 Background

Photometric stereo represents a useful method for recovering high-resolution surface normals from a 3D scene using 2D images taken under $r$ different lighting conditions. One proposed model for the observation process at a single pixel is

$$\boldsymbol{o} = \rho \boldsymbol{Ln} + \boldsymbol{e}, \tag{18}$$

where the $r$ measurments are denoted $\boldsymbol{o} \in \mathbb{R}^r$, $\boldsymbol{n} \in \mathbb{R}^3$ denotes the true 3D surface normal, rows of $\boldsymbol{L} \in \mathbb{R}^{r \times 3}$ define lighting directions, $\rho$ is the diffuse albedo, acting here as a scalar multiplier, and $\boldsymbol{e}$ represents an aggregations of shadows, specular highlights, or other corrupting influences [6, 13]. If $\boldsymbol{e}$ were not present, then the surface normals can be uniquely determined using a simple least-squares fit. However, a more robust alternative involves solving

$$\min_{\tilde{\boldsymbol{n}}, \boldsymbol{e}} \|\boldsymbol{e}\|_0 \quad \text{s.t.} \quad \boldsymbol{o} = \boldsymbol{L}\tilde{\boldsymbol{n}} + \boldsymbol{e}, \tag{19}$$

where $\tilde{\boldsymbol{n}}$ is the surface normal rescaled by $\rho$, which is equivalent to computing [6]

$$\min_{\boldsymbol{e}} \|\boldsymbol{e}\|_0 \quad \text{s.t. } \text{Proj}_{\text{null}[L^\top]}(\boldsymbol{o}) = \text{Proj}_{\text{null}[L^\top]}(\boldsymbol{e}). \tag{20}$$

It can be shown that this formulation has the exact same structure as (M.2) in the limit $\lambda \to 0$, if we assume that $\boldsymbol{y} \triangleq \text{Proj}_{\text{null}[L^\top]}(\boldsymbol{o})$ and $\boldsymbol{\Phi}$ is defined such that $\boldsymbol{\Phi}\boldsymbol{e} = \text{Proj}_{\text{null}[L^\top]}(\boldsymbol{e})$.

## 7.2 Experiment Design

We test algorithms separately on two objects, 'Bunny' and 'Caesar' from [7]. First lighting conditions are generated whose directions are randomly selected from a hemisphere with the object placed at the center. Then 32-bit HDR gray-scale images of the object are rendered with foreground masks and a randomly chosen $\rho$, 0.64 for Bunny and 0.8 for Caesar. The resulting image resolution for Bunny is $(256 \times 256)$ while for Caesar it is $(300 \times 400)$. Given $\boldsymbol{L}$, we apply a singular value decomposition to get $\boldsymbol{\Phi} = \text{Proj}_{\text{null}[L^\top]}$ and the ground truth error vector $\boldsymbol{e}^* = \boldsymbol{o} - \rho\boldsymbol{L}\boldsymbol{n}$.

For training, we have to synthesize candidate sparse errors $\boldsymbol{e}$ since there is no photometric stereo database for this purpose. We adopt the basic pipeline from [14] to accomplish this, which amounts to a form of weakly supervised learning. First we draw a support pattern for $\boldsymbol{e}$ uniformly at random with cardinality $d$ sampled from the range $[d_1, d_2]$. Nonzero values of $\boldsymbol{e}$ are assigned iid random values from $\mathcal{N}(\mu_e, \sigma_e)$. Finally, we can naturally compute observations $\boldsymbol{y} = \boldsymbol{\Phi}\boldsymbol{e}$ which serve as network inputs. Although $d_1, d_1, \mu_e$, and $\sigma_e$ are all tunable, beyond this, no attempt is made to match the true outlier distributions encountered in applications of photometric stereo. After training on synthetic data (weak supervision), we directly apply the resulting model to the gray-scale images without any additional application-specific tuning. During the testing stage, for each surface point, we use our model to approximately solve (20). Since the network outputs a probability map for the outlier support set, we choose $k$ indices with the least probability as inliers and use them to compute $\boldsymbol{n}$ via least squares.

**Hyperparameters:** We conduct experiments under three situations using $r = 10, 20, 40$ images corresponding to $r$ different lighting conditions. As for model capacity, we set the size of hidden states of LSTM cells equal to $2r$. Other training settings remain default as in Section 5.3.

**Visual Results:** See Figure 5.

## 8. Additional Experiments and Self-Comparisons

We first more provide more evaluation details for generic sparse recovery problems, followed by a number of ablation studies.

(a) GT (Bunny)

(b) GT (Caesar)

(c) Ours($r = 10$)  (d) Ours($r = 20$)  (e) Ours($r = 40$)

(f) SBL($r = 10$)  (g) SBL($r = 20$)  (h) SBL($r = 40$)

(i) Ours($r = 10$)  (j) Ours($r = 20$)  (k) Ours($r = 40$)

(l) SBL($r = 10$)  (m) SBL($r = 20$)  (n) SBL($r = 40$)

Figure 5: Photometric stereo reconstruction error maps with different numbers ($r$) of gray-scale images. These correspond with Table (M.1) results.

Table 1: Attributes of our models used in producing Figure M.3(c) results.

| model | Hidden Unit Size | #Parameters | Training Time(sec./epoch) | S-Acc |
|---|---|---|---|---|
| GRU-small | 320 | 1296740 | 98.314 | 0.1588 |
| LSTM-small | 272 | 1213220 | 119.605 | 0.3017 |
| GFGRU-small | 220 | 1285340 | 170.282 | 0.4651 |
| GFLSTM-small | 200 | 1209300 | 172.013 | 0.4691 |
| GRU-big | 680 | 4958660 | 234.024 | 0.3069 |
| LSTM-big | 600 | 5037700 | 312.642 | 0.2637 |
| GFGRU-big | 455 | 4903635 | 318.690 | 0.6028 |
| GFLSTM-big | 425 | 4864650 | 310.447 | 0.6087 |

## 8.1 Further Details for Sparse Vector Recovery Evaluation

Table 1 lists all the important attributes of our self-comparison models from Figure M.3(c) in the main paper. In terms of evaluation on generic problems, we define *strict accuracy*(s-acc) and *loose accuracy*(l-acc) via

$$\mathcal{S}_{gt} = \{j : x_j^* \neq 0\}, \quad \mathcal{S}_{pred}(d) = \{j : p_j \text{ is one of the } d \text{ largest outputs}\} \tag{21}$$

$$\text{s-acc} = \frac{1}{N}\sum_{i=1}^{N} \mathbb{I}\left[\mathcal{S}_{gt}^i = \mathcal{S}_{pred}^i(d)\right], \quad \text{l-acc} = \frac{1}{N}\sum_{i=1}^{N} \frac{|\mathcal{S}_{gt}^i \cap \mathcal{S}_{pred}^i(n)|}{d}, \tag{22}$$

where $N$ is the number of samples.

## 8.2 Ablation Study for Generic Sparse Estimation Problem

In Table 2, we list an ablation results of GFLSTM models with different hyperparameters for the $\frac{d}{n} = 0.4$ case. Enlarging capacity generally benefits the performance especially when the capacity is relatively small. However, the effectiveness and efficiency of changing hidden size, LSTM layers, or number of unrolling steps varies. Stacking too many LSTM layers is the least efficient way the enlarge the model capacity considering the trade-off between training time and performance improvement. As for unrolling, insufficient steps (for example, under 10) can impair model performance while excessive unrolling is a waste of computation. And hidden layer size is a quite effective way to control the model capacity.

## 9. Technical Proofs

Here we present proofs of our technical propositions.

**Proof of Proposition M.1**

It has been demonstrated in [11] that using

$$w_i^{(t+1)} = \left[\phi_i^\top \left(\lambda \boldsymbol{I} + \boldsymbol{\Phi}\boldsymbol{\Gamma}^{(t)}\boldsymbol{\Phi}^\top\right)^{-1} \phi_i\right]^{\frac{1}{2}} \tag{23}$$

Table 2: Results from models with different capacities. There are three main capacity control factors: number of hidden units, number of LSTM stacked layers, and number of LSTM unrolling steps. For various capacities settings, the total number of parameters, training time per epoch (sec.), and strict-accuracy result are listed.

| #Hidden | #Layers | #Unroll | #Parameters | Time | S-Acc |
|---------|---------|---------|-------------|------|-------|
| 200 | 2 | 5 | 1089300 | 99.333 | 0.0524 |
| 200 | 2 | 8 | 1149300 | 131.723 | 0.2211 |
| 200 | 2 | 11 | 1209300 | 172.013 | 0.4691 |
| 200 | 2 | 14 | 1269300 | 206.084 | 0.4707 |
| 200 | 2 | 17 | 1329300 | 240.847 | 0.5060 |
| 200 | 3 | 5 | 2497700 | 160.404 | 0.1455 |
| 200 | 3 | 8 | 2557700 | 234.113 | 0.4146 |
| 200 | 3 | 11 | 2617700 | 309.859 | 0.4776 |
| 200 | 3 | 14 | 2677700 | 383.898 | 0.5319 |
| 200 | 3 | 17 | 2737700 | 446.197 | 0.6011 |
| 200 | 4 | 5 | 4787300 | 254.694 | 0.2561 |
| 200 | 4 | 8 | 4847300 | 380.841 | 0.5619 |
| 200 | 4 | 11 | 4907300 | 517.010 | 0.5802 |
| 200 | 4 | 14 | 4967300 | 631.771 | 0.6046 |
| 200 | 4 | 17 | 5027300 | 764.149 | 0.6156 |
| 425 | 2 | 5 | 4609650 | 173.272 | 0.0976 |
| 425 | 2 | 8 | 4737150 | 235.728 | 0.4334 |
| 425 | 2 | 11 | 4864650 | 312.642 | 0.6087 |
| 425 | 2 | 14 | 4992150 | 378.839 | 0.6595 |
| 425 | 2 | 17 | 5119650 | 442.985 | 0.6697 |
| 425 | 3 | 5 | 10949375 | 316.927 | 0.2598 |
| 425 | 3 | 8 | 11076875 | 470.105 | 0.6043 |
| 425 | 3 | 11 | 11204375 | 616.227 | 0.6584 |
| 425 | 3 | 14 | 11331875 | 763.941 | 0.6359 |
| 425 | 3 | 17 | 11459375 | 927.643 | 0.6427 |
| 425 | 4 | 5 | 21265400 | 540.034 | 0.3653 |
| 425 | 4 | 8 | 21392900 | 821.973 | 0.6376 |
| 425 | 4 | 11 | 21520400 | 1096.844 | 0.6372 |
| 425 | 4 | 14 | 21647900 | 1389.964 | 0.6569 |
| 425 | 4 | 17 | 21775400 | 1655.417 | 0.6618 |
| 600 | 2 | 11 | 9387700 | 461.414 | 0.6494 |
| 600 | 2 | 14 | 9567700 | 568.918 | 0.6795 |
| 600 | 2 | 17 | 9747700 | 698.721 | 0.6682 |

will satisfy the stated conditions of Proposition M.1. Now assume that $\mathbf{\Gamma}^{(t)}$ is full rank or invertible, i.e., $\gamma_j^{(t)} > 0$ for all $j$. Using the matrix inversion lemma, we have

$$\boldsymbol{\phi}_i^\top \left( \lambda \mathbf{I} + \mathbf{\Phi}\mathbf{\Gamma}^{(t)}\mathbf{\Phi}^\top \right)^{-1} \boldsymbol{\phi}_i = \tfrac{1}{\lambda}\boldsymbol{\phi}_i^\top \left( \mathbf{I} - \tfrac{1}{\lambda}\mathbf{\Phi} \left[ \left( \mathbf{\Gamma}^{(t)} \right)^{-1} + \tfrac{1}{\lambda}\mathbf{\Phi}^\top\mathbf{\Phi} \right]^{-1} \mathbf{\Phi}^\top \right) \boldsymbol{\phi}_i. \qquad (24)$$

Given that the matrix inverse is a convex function, and that additive translations preserve convexity, it follows that $\quad \tfrac{1}{\lambda^2}\boldsymbol{\phi}_i^\top\mathbf{\Phi} \left[ \left( \mathbf{\Gamma}^{(t)} \right)^{-1} + \tfrac{1}{\lambda}\mathbf{\Phi}^\top\mathbf{\Phi} \right]^{-1} \mathbf{\Phi}^\top\boldsymbol{\phi}_i \quad$ is a convex function of $\left( \mathbf{\Gamma}^{(t)} \right)^{-1}$. Therefore the negation of this term is concave, and so overall (24) is a concave function of $\left( \mathbf{\Gamma}^{(t)} \right)^{-1}$. This then implies that we can express (24) as a minimization of upper-bounding hyperplanes via

$$\boldsymbol{\phi}_i^\top \left( \lambda \mathbf{I} + \mathbf{\Phi}\mathbf{\Gamma}^{(t)}\mathbf{\Phi}^\top \right)^{-1} \boldsymbol{\phi}_i \;=\; \min_{\boldsymbol{z}} g(\boldsymbol{z}) + \sum_{j=1}^m \frac{f(z_j)}{\gamma_i} \qquad (25)$$

for some functions $f$ and $g$ and variational parameters $\boldsymbol{z} = [z_1, \ldots, z_m]^\top$. Such a decomposition is not unique; however, using linear algebraic manipulations, it can be easily verified that

$$\boldsymbol{\phi}_i^\top \left( \lambda \mathbf{I} + \mathbf{\Phi}\mathbf{\Gamma}^{(t)}\mathbf{\Phi}^\top \right)^{-1} \boldsymbol{\phi}_i \;=\; \min_{\boldsymbol{z}} \frac{1}{\lambda}\|\boldsymbol{\phi}_i - \mathbf{\Phi}\boldsymbol{z}\|_2^2 + \sum_{j=1}^m \frac{z_j^2}{\gamma_j^{(t)}} \qquad (26)$$

is one such viable representation.

To handle the more general case where some $\gamma_j^{(t)} = 0$, we use $\bar{\mathbf{\Phi}}$ to denote the columns $\boldsymbol{\phi}_j$ such that $j \in \mathrm{supp}[\boldsymbol{\gamma}]$, and likewise $\bar{\mathbf{\Gamma}}^{(t)}$ and $\bar{\boldsymbol{z}}$ the corresponding submatrix of $\mathbf{\Gamma}^{(t)}$ and elements of $\boldsymbol{z}$ respectively. It then naturally follows that

$$
\begin{aligned}
\boldsymbol{\phi}_i^\top \left( \lambda \mathbf{I} + \mathbf{\Phi}\mathbf{\Gamma}^{(t)}\mathbf{\Phi}^\top \right)^{-1} \boldsymbol{\phi}_i \;&=\; \boldsymbol{\phi}_i^\top \left( \lambda \mathbf{I} + \bar{\mathbf{\Phi}}\bar{\mathbf{\Gamma}}^{(t)}\bar{\mathbf{\Phi}}^\top \right)^{-1} \boldsymbol{\phi}_i \\
&=\; \min_{\bar{\boldsymbol{z}}} \frac{1}{\lambda}\|\boldsymbol{\phi}_i - \bar{\mathbf{\Phi}}\bar{\boldsymbol{z}}\|_2^2 + \sum_{j=1}^{\|\boldsymbol{\gamma}^{(t)}\|_0} \frac{\bar{z}_j^2}{\bar{\gamma}_j^{(t)}} \\
&=\; \min_{\boldsymbol{z}:\mathrm{supp}[\boldsymbol{z}]\subseteq\mathrm{supp}[\boldsymbol{\gamma}^{(t)}]} \frac{1}{\lambda}\|\boldsymbol{\phi}_i - \mathbf{\Phi}\boldsymbol{z}\|_2^2 + \sum_{j\in\mathrm{supp}[\boldsymbol{\gamma}^{(t)}]} \frac{z_j^2}{\gamma_j^{(t)}}.
\end{aligned}
\qquad (27)
$$

**Proof of Proposition M.3**

The original SBL objective is given by

$$\mathcal{L}(\boldsymbol{\gamma}) = \boldsymbol{y}^\top \left( \mathbf{\Phi}\mathbf{\Gamma}\mathbf{\Phi}^\top + \lambda \mathbf{I} \right)^{-1} \boldsymbol{y} + \log \left| \mathbf{\Phi}\mathbf{\Gamma}\mathbf{\Phi}^\top + \lambda \mathbf{I} \right|, \qquad (28)$$

where the first term is convex in $\boldsymbol{\gamma}$ while the second is concave, ultimately resulting in a non-convex function. For optimization purposes, it is convenient to decouple elements of

$\boldsymbol{\gamma}$ via a series of upper bounds, the iterative minimization of which leads to LSTM-like updates given judicious choices for these bounds.

To begin, we have the linear upper bound

$$h(\boldsymbol{\gamma}) \triangleq \log \left| \boldsymbol{\Phi} \boldsymbol{\Gamma} \boldsymbol{\Phi}^\top + \lambda \boldsymbol{I} \right| \leq h(\widetilde{\boldsymbol{\gamma}}) + (\boldsymbol{\gamma} - \widetilde{\boldsymbol{\gamma}})^\top \nabla h(\widetilde{\boldsymbol{\gamma}}), \tag{29}$$

which is always realizable for any $\widetilde{\boldsymbol{\gamma}} \in \mathbb{R}_+^m$ given the concavity of $h(\boldsymbol{\gamma})$ [2]. This bound decouples individual elements of $\boldsymbol{\gamma}$ into a linear summation that facilitates convenient, separable optimization. Analogously, for the data-dependent term we have

$$\boldsymbol{y}^\top \left( \boldsymbol{\Phi} \boldsymbol{\Gamma} \boldsymbol{\Phi}^\top + \lambda \boldsymbol{I} \right)^{-1} \boldsymbol{y} \leq \tfrac{1}{\lambda} \|\boldsymbol{y} - \boldsymbol{\Phi} \boldsymbol{u}\|_2^2 + \boldsymbol{u}^\top \boldsymbol{\Gamma}^{-1} \boldsymbol{u}. \tag{30}$$

This bound holds for all $\boldsymbol{u} \in \mathbb{R}^m$, with equality when $\boldsymbol{u} = \boldsymbol{\Gamma} \boldsymbol{\Phi}^\top \left( \lambda \boldsymbol{I} + \boldsymbol{\Phi} \boldsymbol{\Gamma} \boldsymbol{\Phi}^\top \right)^{-1} \boldsymbol{y}$ [12].[2] Although the r.h.s. of (30) has effectively decoupled $\boldsymbol{\gamma}$ (given that $\boldsymbol{\Gamma}$ is diagonal, $\boldsymbol{u}^\top \boldsymbol{\Gamma}^{-1} \boldsymbol{u}$ is separable), it has introduced new auxiliary variables $\boldsymbol{u}$ which are inter-mixed via a $\boldsymbol{\Phi}$-dependent norm. However, we can further bound this term using

$$f(\boldsymbol{u}) \triangleq \tfrac{1}{\lambda} \|\boldsymbol{y} - \boldsymbol{\Phi} \boldsymbol{u}\|_2^2 \leq f(\widetilde{\boldsymbol{u}}) + (\boldsymbol{u} - \widetilde{\boldsymbol{u}})^\top \nabla f(\widetilde{\boldsymbol{u}}) + \tfrac{1}{2\mu} \|\boldsymbol{u} - \widetilde{\boldsymbol{u}}\|_2^2, \tag{31}$$

for any $\widetilde{\boldsymbol{u}} \in \mathbb{R}^m$ provided that $\mu \in \left( 0, \lambda / \left\| \boldsymbol{\Phi}^\top \boldsymbol{\Phi} \right\| \right]$. This occurs because $\nabla f(\boldsymbol{u})$ is Lipschitz continuous with Lipschitz constant $\tfrac{1}{\lambda} \left\| \boldsymbol{\Phi}^\top \boldsymbol{\Phi} \right\|$, in which case a quadratic upper bound can always be constructed as in (31).

Combining terms, we arrive at the auxiliary objective function

$$\mathcal{L}(\boldsymbol{\gamma}, \widetilde{\boldsymbol{\gamma}}, \boldsymbol{u}, \widetilde{\boldsymbol{u}}) \triangleq h(\widetilde{\boldsymbol{\gamma}}) + (\boldsymbol{\gamma} - \widetilde{\boldsymbol{\gamma}})^\top \nabla h(\widetilde{\boldsymbol{\gamma}}) + \boldsymbol{u}^\top \boldsymbol{\Gamma}^{-1} \boldsymbol{u} + f(\widetilde{\boldsymbol{u}}) + (\boldsymbol{u} - \widetilde{\boldsymbol{u}})^\top \nabla f(\widetilde{\boldsymbol{u}}) + \tfrac{1}{2\mu} \|\boldsymbol{u} - \widetilde{\boldsymbol{u}}\|_2^2, \tag{32}$$

where $\widetilde{\boldsymbol{\gamma}}$, $\boldsymbol{u}$, and $\widetilde{\boldsymbol{u}}$ can be viewed in this context as additional latent variables, sometimes referred to as variational paramters. And by design, for any $\boldsymbol{\gamma}$ we have that

$$\mathcal{L}(\boldsymbol{\gamma}) = \min_{\widetilde{\boldsymbol{\gamma}}, \boldsymbol{u}, \widetilde{\boldsymbol{u}}} \mathcal{L}(\boldsymbol{\gamma}, \widetilde{\boldsymbol{\gamma}}, \boldsymbol{u}, \widetilde{\boldsymbol{u}}) \leq \mathcal{L}(\boldsymbol{\gamma}, \widetilde{\boldsymbol{\gamma}}, \boldsymbol{u}, \widetilde{\boldsymbol{u}}). \tag{33}$$

Additionally, this minimization can be accomplished exactly using the stated updates from Section M.2.3. The details are as follows.

Assume that we would like to reduce $\mathcal{L}(\boldsymbol{\gamma})$ starting from some arbitrary point $\boldsymbol{\gamma}^{(t)}$. If we choose

$$\widetilde{\boldsymbol{\gamma}}^{(t)} = \boldsymbol{\gamma}^{(t)}, \qquad \boldsymbol{u}^{(t)} = \boldsymbol{\Gamma}^{(t)} \boldsymbol{\Phi}^\top \left( \lambda \boldsymbol{I} + \boldsymbol{\Phi} \boldsymbol{\Gamma}^{(t)} \boldsymbol{\Phi}^\top \right)^{-1} \boldsymbol{y}, \qquad \widetilde{\boldsymbol{u}}^{(t)} = \boldsymbol{u}^{(t)}, \tag{34}$$

then $\mathcal{L}\left(\boldsymbol{\gamma}^{(t)}\right) = \mathcal{L}\left(\boldsymbol{\gamma}^{(t)}, \widetilde{\boldsymbol{\gamma}}^{(t)}, \boldsymbol{u}^{(t)}, \widetilde{\boldsymbol{u}}^{(t)}\right)$ by construction, i.e., these values simultaneously optimize $\mathcal{L}(\boldsymbol{\gamma}^{(t)}, \widetilde{\boldsymbol{\gamma}}, \boldsymbol{u}, \widetilde{\boldsymbol{u}}) \geq \mathcal{L}\left(\boldsymbol{\gamma}^{(t)}\right)$ per our structuring of the respective bounds. Our strategy will now be to solve

$$\min_{\boldsymbol{\gamma}, \boldsymbol{u}} \mathcal{L}\left(\boldsymbol{\gamma}, \widetilde{\boldsymbol{\gamma}}^{(t)}, \boldsymbol{u}, \widetilde{\boldsymbol{u}}^{(t)}\right) \tag{35}$$

---

2. Additionally, if some $\boldsymbol{\gamma}_j = 0$ while $u_j \neq 0$, we simply define this bound to be infinity. All subsequent update rules are well-defined regardless.

in closed form in order to obtain a new $\gamma^{(t+1)}$ that reduces to the original objective function $\mathcal{L}(\gamma)$. For this purpose we define $w^{(t)}$ such that

$$\left(w^{(t)}\right)^2 = \nabla h\left(\widetilde{\gamma}^{(t)}\right) = \mathrm{diag}\left[\Phi^\top\left(\lambda I + \Phi \Gamma^{(t)} \Phi^\top\right)^{-1} \Phi\right], \qquad (36)$$

where the squaring operator is applied element-wise and the gradient is calculated using standard formulae. Note that this representation is always possible given that $\nabla h(\widetilde{\gamma})$ must have non-negative elements since $h$ is a non-decreasing, concave function.

By excluding irrelevant terms, taking derivatives, and equating to zero, it follows that

$$\left(w^{(t)}\right)^{-1} \odot |u| \;=\; \arg\min_{\gamma} \mathcal{L}\left(\gamma, \widetilde{\gamma}^{(t)}, u, \widetilde{u}^{(t)}\right) \;\equiv\; \arg\min_{\gamma} \sum_i \left[\left(w_i^{(t)}\right)^2 \gamma_i^{(t)} + \frac{u_i^2}{\gamma_i^{(t)}}\right]. \quad (37)$$

Plugging this value into the $\gamma$-dependent terms from $\mathcal{L}\left(\gamma, \widetilde{\gamma}^{(t)}, u, \widetilde{u}^{(t)}\right)$, we find that

$$\gamma^\top \nabla h(\widetilde{\gamma}) + u^\top \Gamma^{-1} u \;\equiv\; 2 w^{(t)} \odot |u|. \qquad (38)$$

Therefore, a conditionally optimal version of $u$ can be achieved by solving

$$
\begin{aligned}
(u^*)^{(t)} &\triangleq \arg\min_{u} \; \mathcal{L}\left(\left[w^{(t)}\right]^{-1} \odot |u|, \widetilde{\gamma}^{(t)}, u, \widetilde{u}^{(t)}\right) \\
&\equiv \arg\min_{u} \; 2 w^{(t)} \odot |u| + u^\top \nabla f\left(\widetilde{u}^{(t)}\right) + \tfrac{1}{2\mu} \left\|u - \widetilde{u}^{(t)}\right\|_2^2 \\
&\equiv \arg\min_{u} \; 2 w^{(t)} \odot |u| + \tfrac{1}{2\mu} \left\|u - \left[\widetilde{u}^{(t)} - \mu \nabla f\left(\widetilde{u}^{(t)}\right)\right]\right\|_2^2. \qquad (39)
\end{aligned}
$$

This expression can be optimized independently across each $u_i$, leading to

$$
\begin{aligned}
(u_i^*)^{(t)} &= S_{2\lambda w_i^{(t)}}\left(\widetilde{u}_i^{(t)} - \mu\left[\nabla f\left(\widetilde{u}^{(t)}\right)\right]_i\right) \\
&= S_{2\lambda w_i^{(t)}}\left(\widetilde{u}_i^{(t)} + \mu\left[\Phi^\top\left(y - \Phi\widetilde{u}^{(t)}\right)\right]_i\right) \qquad (40)
\end{aligned}
$$

where $S_\omega$ is a soft threshold operator. Moreover, based on (37), it follows that

$$\gamma^{(t+1)} = \left(w^{(t)}\right)^{-1} \odot \left|(u^*)^{(t)}\right| \qquad (41)$$

will be such that

$$\mathcal{L}\left(\gamma^{(t+1)}\right) \leq \mathcal{L}\left(\gamma^{(t+1)}, \widetilde{\gamma}^{(t)}, (u^*)^{(t)}, \widetilde{u}^{(t)}\right) \leq \mathcal{L}\left(\gamma^{(t)}, \widetilde{\gamma}^{(t)}, u^{(t)}, \widetilde{u}^{(t)}\right) = \mathcal{L}\left(\gamma^{(t)}\right). \qquad (42)$$

Therefore, by following the above process, $\mathcal{L}(\gamma)$ will be reduced (or left unchanged). One attractive feature of this formulation is that $\gamma$ can be optimized jointly with at least one set of variational parameters (in this case $u$), as opposed to most majorization-minimization strategies [5] that fix the upper bound before minimizing the original variables (in this case $\gamma$).

If we choose $\alpha(\gamma) = 1$ and $\beta(\gamma) = 0$, then these steps exactly mirror the revised SBL iterations from Section M.2.3 once we define $x^{(t+1)} \triangleq (u^*)^{(t)}$ and note that $\sigma_{in}^{(t+1)} \odot \bar{x}^{(t+1)}$

is tantamount to soft-thresholding. Demonstrating the more general case involves a few additional manipulations.

Following the updates described above, we have

$$
\begin{aligned}
\mathcal{L}\left(\boldsymbol{\gamma}^{(t)}\right) &= \mathcal{L}\left(\boldsymbol{\gamma}^{(t)}, \widetilde{\boldsymbol{\gamma}}^{(t)}, \boldsymbol{u}^{(t)}, \widetilde{\boldsymbol{u}}^{(t)}\right) \\
&= h\left(\widetilde{\boldsymbol{\gamma}}^{(t)}\right) + \left(\boldsymbol{u}^{(t)}\right)^{\top}\left(\boldsymbol{\Gamma}^{(t)}\right)^{-1}\boldsymbol{u}^{(t)} + \tfrac{1}{\lambda}\left\|\boldsymbol{y} - \boldsymbol{\Phi}\boldsymbol{u}^{(t)}\right\|_2^2 \\
&\geq h\left(\widetilde{\boldsymbol{\gamma}}^{(t)}\right) - \left(\widetilde{\boldsymbol{\gamma}}^{(t)}\right)^{\top}\nabla h\left(\widetilde{\boldsymbol{\gamma}}^{(t)}\right) + 2\boldsymbol{w}^{(t)}\odot\left|\boldsymbol{u}^{(t)}\right| + \tfrac{1}{\lambda}\left\|\boldsymbol{y} - \boldsymbol{\Phi}\boldsymbol{u}^{(t)}\right\|_2^2 \\
&= h\left(\widetilde{\boldsymbol{\gamma}}^{(t)}\right) - \left(\widetilde{\boldsymbol{\gamma}}^{(t)}\right)^{\top}\nabla h\left(\widetilde{\boldsymbol{\gamma}}^{(t)}\right) + 2\boldsymbol{w}^{(t)}\odot\left|\boldsymbol{u}^{(t)}\right| + f\left(\widetilde{\boldsymbol{u}}^{(t)}\right) \\
&\quad + \left(\boldsymbol{u}^{(t)} - \widetilde{\boldsymbol{u}}^{(t)}\right)^{\top}\nabla f\left(\widetilde{\boldsymbol{u}}^{(t)}\right) + \tfrac{1}{2\mu}\left\|\boldsymbol{u}^{(t)} - \widetilde{\boldsymbol{u}}^{(t)}\right\|_2^2
\end{aligned}
\tag{43}
$$

given that presently $\boldsymbol{u}^{(t)} = \widetilde{\boldsymbol{u}}^{(t)}$. Previously we optimized this expression with respect to $\boldsymbol{u}$ and obtained the soft-threshold estimator $(\boldsymbol{u}^{*})^{(t)}$. However, suppose we instead evaluate at an alternative point $(\boldsymbol{u}')^{(t)}$ defined recursively such that

$$
\left(\boldsymbol{u}'\right)^{(t)} \equiv \boldsymbol{x}^{(t+1)} = \boldsymbol{\beta}\left(\boldsymbol{\gamma}^{(t)}\right)\odot\boldsymbol{x}^{(t)} + \boldsymbol{\alpha}\left(\boldsymbol{\gamma}^{(t)}\right)\odot(\boldsymbol{u}^{*})^{(t)}.
\tag{44}
$$

Then finally we have

$$
\begin{aligned}
\mathcal{L}\left(\boldsymbol{\gamma}^{(t)}\right) &\geq h\left(\widetilde{\boldsymbol{\gamma}}^{(t)}\right) - \left(\widetilde{\boldsymbol{\gamma}}^{(t)}\right)^{\top}\nabla h\left(\widetilde{\boldsymbol{\gamma}}^{(t)}\right) + 2\boldsymbol{w}^{(t)}\odot\left|\boldsymbol{u}^{(t)}\right| + \tfrac{1}{\lambda}\left\|\boldsymbol{y} - \boldsymbol{\Phi}\boldsymbol{u}^{(t)}\right\|_2^2 \\
&\geq h\left(\widetilde{\boldsymbol{\gamma}}^{(t)}\right) - \left(\widetilde{\boldsymbol{\gamma}}^{(t)}\right)^{\top}\nabla h\left(\widetilde{\boldsymbol{\gamma}}^{(t)}\right) + 2\boldsymbol{w}^{(t)}\odot\left|(\boldsymbol{u}')^{(t)}\right| + \tfrac{1}{\lambda}\left\|\boldsymbol{y} - \boldsymbol{\Phi}(\boldsymbol{u}')^{(t)}\right\|_2^2 \\
&= h\left(\widetilde{\boldsymbol{\gamma}}^{(t)}\right) + \left(\boldsymbol{\gamma}^{(t+1)} - \widetilde{\boldsymbol{\gamma}}^{(t)}\right)^{\top}\nabla h\left(\widetilde{\boldsymbol{\gamma}}^{(t)}\right) + \left(\boldsymbol{x}^{(t+1)}\right)^{\top}\left(\boldsymbol{\Gamma}^{(t+1)}\right)^{-1}\boldsymbol{x}^{(t+1)} \\
&\quad + \tfrac{1}{\lambda}\left\|\boldsymbol{y} - \boldsymbol{\Phi}\boldsymbol{x}^{(t+1)}\right\|_2^2 \\
&\geq \mathcal{L}\left(\boldsymbol{\gamma}^{(t+1)}\right),
\end{aligned}
\tag{45}
$$

where now $\boldsymbol{\gamma}^{(t+1)} = \left(\boldsymbol{w}^{(t)}\right)^{-1}\odot\left|(\boldsymbol{u}')^{(t)}\right|$. The first inequality follows from (43), the second from the monotone cell update property, and the third via the original construction of the majorization-minimization algorithm. This process then exactly mirrors the iterations from Section M.2.3, with guaranteed cost function descent.