[Reviews · NeurIPS 2017]

Reviewer 1



This paper proposes a recurrent network for sparse estimation inspired on sparse Bayesian learning (SBL). It first shows that a recurrent architecture can implement a variant of SBL, showing through simulations that different quantities have different time dynamics, which motivates leveraging ideas from recurrent-network design to adapt the architecture. This leads to a recurrent network that seems to outperform approaches based on optimization in simulations and two applications. Strengths: The idea of learning a recurrent network for sparse estimation has great potential impact. The paper is very well written. The authors motivate their design decisions in detail and report numerical experiments that are quite thorough and indicate that the technique is successful for challenging sparse-decomposition problems. Weaknesses: A lot of the details about the implementation of the method and the experiments are deferred to the supplementary material, so that the main paper is a bit vague. I find this understandable due to length limitations.

Reviewer 2



The authors explored a connection between Bayesian sparsity and LSTM networks and then extended the work to gated feedback networks. Specifically, the authors first discussed the relationship between sparse Bayesian learning and iterative reweighted l1 regularization, then discussed the relationship with LSTM, and finally extended to gated feedback networks. Experimental results on synthetic and real data sets were reported.

Reviewer 3



The paper presents an idea of casting the sparse Bayesian learning as a recurrent neural network structure, which enables the learning of the functions without having to hand-craft the iterations. The paper is well written and clearly presented. The presented idea is interesting and aligns with some recent works presented in the literature on establishing links between sparse representation and deep neural networks, such as sparse encoder, LISTA, etc. The experiments on DOA estimation and 3D geometry reconstruction are interesting and show the diversity of the potential applications of this technique.